# Resilient cooperators stabilize long-run cooperation in the finitely repeated Prisoner's Dilemma

Andrew Mao[1], Lili Dworkin[2], Siddharth Suri[1] & Duncan J. Watts[1]

Learning in finitely repeated games of cooperation remains poorly understood in part because their dynamics play out over a timescale exceeding that of traditional lab experiments. Here, we report results of a virtual lab experiment in which 94 subjects play up to 400 ten-round games of Prisoner's Dilemma over the course of twenty consecutive weekdays. Consistent with previous work, the typical round of first defection moves earlier for several days; however, this unravelling process stabilizes after roughly one week. Analysing individual strategies, we find that approximately 40% of players behave as resilient cooperators who avoid unravelling even at significant cost to themselves. Finally, using a standard learning model we predict that a sufficiently large minority of resilient cooperators can permanently stabilize unravelling among a majority of rational players. These results shed hopeful light on the long-term dynamics of cooperation, and demonstrate the importance of long-run experiments.

[1] Microsoft Research, 641 Avenue of the Americas, 7th Floor, New York, New York 10011, USA. [2] Socratic.org, 1133 Broadway, Suite 423, New York, New York 10010, USA. Correspondence and requests for materials should be addressed to A.M. (email: mao@microsoft.com) or to S.S. (email: suri@microsoft.com) or to D.J.W. (email: duncan@microsoft.com).

The mechanisms by which cooperation among humans has evolved and can be sustained have long been of interest to researchers across several disciplines including economics[1–10], sociology[11–14], psychology[15–17], political science[18–20], evolutionary biology[21–26] and complex systems research[27–30]. Despite this extraordinary level of attention, numerous questions remain unresolved. In this paper, we focus on one such question: what happens to cooperation in finitely repeated games when individuals within the same population repeatedly play these games over long intervals of time? Do they begin to exploit one another leading to the eventual erosion of cooperation? Or do they instead remain resilient in the face of occasional exploitation and continue to cooperate even when it is costly to them? Prior work on this question has reached mixed conclusions[1–3,10,31] in large part because the learning dynamics in question plays out over a longer timescale than be accommodated in traditional lab settings.

In a seminal contribution, Kreps et al.[1] proved that a population of entirely rational players could exhibit potentially high levels of cooperation in a finitely repeated game of Prisoner's Dilemma as long as they believe with sufficiently high probability that others will cooperate until they are defected on, and that once defected on, they will retaliate by defecting themselves. Under these conditions, Kreps et al.[1] showed that it would be optimal for a rational player to cooperate on the grounds that the gains to mutual cooperation exceeded the one-time gain of exploiting a conditional cooperator. For the same reason, however, they also showed that self-interest would cause rational players to eventually defect as the end-game approached. This 'rational cooperation' hypothesis provides an elegant explanation of experimental observations of single finitely repeated games, which consistently show high initial cooperation followed by a sharp decrease in the final rounds; however, it leaves unanswered how cooperation will evolve as the repeated game is itself repeated many times. On the one hand, if all players are in fact rational then cooperation should systematically 'unravel' as rational players, increasingly anticipating other rational players, begin to defect on ever earlier rounds. In the long run, all players will defect on all rounds just as predicted by classical backward induction arguments[32], albeit for somewhat different reasons. On the other hand, if sufficiently many players are in fact conditional cooperators, either because they harbour altruistic (that is, other regarding) preferences or for some other reason (for example, they have internalized social norms of fairness[8,12,13,16]), then it will continue to be in the interests of the rational players to cooperate also. In that event, it has been suggested that cooperation could be sustained indefinitely[3].

If the rational cooperation hypothesis is correct, then the long-run fate of cooperation reduces to an empirical question about the frequency of 'true' conditional cooperators and their resilience to occasional exploitation by rational players. It is unclear from the theory alone, however, how many cooperators are needed to sustain cooperation indefinitely, how resilient they must be to exploitation, or what the relationship is between the frequency of conditional cooperators and the level of cooperation among rational players. Empirical work has also failed to answer these questions conclusively, but has generally leaned toward unravelling as the likely long-run outcome[10,31]. For example, although a consistent finding is that somewhere between 40 and 60% of participants cooperate in one-shot dilemmas[4,8,20,28,29], these same studies have been generally pessimistic about the ability of cooperators to resist unravelling in repeated games with rational players. Meanwhile, experiments that have been designed to test for unravelling directly have reached inconsistent conclusions, with some seeming to suggest that unravelling will prevail[2] and others suggesting the opposite[3]. A recent study[10] attributes these

| | C | D |
|---|---|---|
| C | 5,5 | 1,7 |
| D | 7,1 | 3,3 |

**Table 1 | Per round payoff for (Row Player, Column Player).**

inconsistencies largely to different choices of parameters and experimental conditions, and attempts to resolve them with a carefully designed experiment in which over 200 students played between 20 and 30 finitely repeated games of prisoner's dilemma. The authors note a consistent pattern, also present in other studies. First, players tend to converge on one of a number of 'threshold rules,' according to which they cooperate conditionally up until some predetermined round, after which they defect unilaterally. Second, having adopted such a strategy, the players' thresholds creep slowly backward with experience, consistent with unravelling. Extrapolating from these initial trends, the authors conclude that on a sufficiently long timescale cooperation will eventually unravel all the way to zero.

Importantly, however, the authors also note that 'the process is slow enough that … it is not plausible to observe cooperation rates decline to negligible levels in an amount of time that is reasonable to spend in a laboratory.' Here, we address this discrepancy between timescales—of learning dynamics on the one hand and lab experiments on the other hand—by exploiting a novel property of 'virtual labs,' namely that they allow us to bring the same group of subjects back to the 'lab' for many days in succession, and therefore to observe how their behaviour unfolds on much longer timescales than has been possible previously. Analysing results from a Prisoner's Dilemma experiment that ran for nearly a month of real time, we find that a majority of players do indeed seek to exploit one another, and that as a result, cooperation erodes during the first week of play. After that time, however, we find that a significant minority of players—roughly 40%—continue to cooperate as long as their partner cooperates, persistently declining to defect first in spite of being exploited by the majority. Finally, we find that this minority of 'resilient cooperators' has a beneficial effect for the whole population, effectively stabilizing the erosion of cooperation after a period of several days, thereby allowing cooperation to be sustained at a surprisingly high level even among the non-resilient majority.

## Results

**Experimental design**. Our experiment was designed to closely resemble a number of previous studies of finitely repeated Prisoner's Dilemma (PD)[2,3,10]. Anonymous individuals were randomly paired to play a series of ten-round repeated games of PD, where in each round each player was required to choose one of two actions—cooperate (C) or defect (D)—after which they received a payoff from the payoff matrix displayed in Table 1 (see also Supplementary Fig. 1 for screenshots). We note that the payoffs were chosen to satisfy the usual PD inequalities $(T=7) > (R=5) > (P=3) > (S=1)$ and $2R > T+S$; moreover, they were chosen to correspond to the normalized quantities $g = \frac{(T-R)}{(R-P)} = 1$ and $l = \frac{(P-S)}{(R-P)} = 1$, which are toward the low end of the normal range for previous studies[2,3,10,33–36]. After each round, both players were shown the action of the other player, and each could see their own payoff as well as cumulative payoffs up to that game and for the entire experiment (see Supplementary Fig. 1). After each ten-round game players entered a virtual waiting room until all other games had completed (a counter informed players how many others were also waiting), at which

point they were randomly reassigned to new partners and a new set of games commenced. This process was repeated 20 times over the course of a single session, where we again emphasize that players remained anonymous and unidentifiable throughout (see Supplementary Fig. 2 for a visual representation of a single day).

Our experiment's main point of departure from previous work was that rather than conducting our experiment for a single session we retained the same population of subjects for 20 such sessions, held at the same time on consecutive weekdays over the period 4 August – 31 August 2015. The experiment commenced with 113 subjects recruited in advance from Amazon's Mechanical Turk. To minimize latency in the user interface and language barriers in delivering instructions, we restricted participation to residents of the US and Canada; however, the subject pool was otherwise diverse with respect to location (31 US states), age (18–61) and gender (47% female) (see Supplementary Figs 3 and 4 for more details of the player population). Also to minimize latency, we split the population into two sessions held each day at 13:00 hours EDT ($n = 56$) and 15:00 hours EDT ($n = 57$), respectively. Players were assigned randomly to a session at the outset of the experiment and were retained in that session for the duration of the experiment. Although there were some slight differences between the two sessions, behaviour—including attrition—was qualitatively indistinguishable, thus for all results stated in the main text we treat the two sessions as a single population (noting that pooling of subjects from multiple experimental sessions is a common practice in traditional lab

experiments). Sessions lasted an average of 35 min and players were paid in proportion to their cumulative payoff. Players earned an average of $4.47 per session corresponding to an hourly wage of ~$7.66, substantially higher than the self-reported average wage for tasks on Mechanical Turk[37]. To minimize attrition, we also offered an additional one-time bonus of $20 for completing at least 18 of the 20 sessions, payable at the end of the experiment. Subjects who missed more than two sessions were excluded from the experiment and prevented from completing any remaining sessions, thereby forfeiting the bonus along with any unearned compensation. Of the initial population 94 subjects (83%) satisfied our completion criterion, earning an average variable compensation of $87.03 and $107.03 in total (we found no significant differences between dropouts and non-dropouts; see 'Methods' section for more details of recruiting and attrition). Over the course of the experiment these subjects played an average of 375 ten-round games each, making 3,720 individual decisions each for a total of 374,251 decisions collectively (see Supplementary Fig. 5 for a visual representation of the entire experiment).

**Initial cooperation and unravelling.** Figure 1a shows cooperation levels in rounds 1 (green), 8 (blue), 9 (purple) and 10 (red) over the course of the experiment. On day 1 the first round cooperation rates started at over 80%, a figure that is not unprecedented among previous studies[38], but is substantially

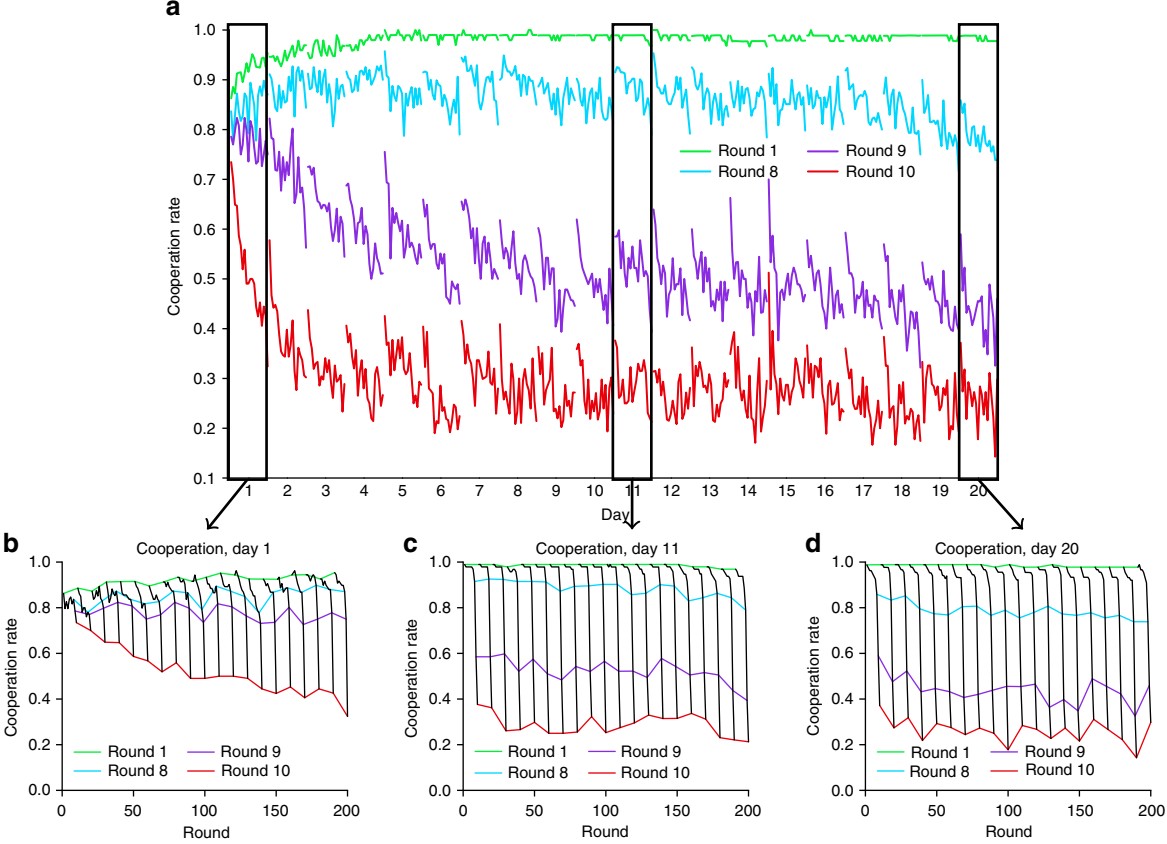

**Figure 1 | Cooperation over time.** (**a**) Average cooperation rate for rounds 1 (green), 8 (blue), 9 (purple) and 10 (red) as a function of time over the 400 games of the experiment. The experiment ran for 20 consecutive weekdays, each of which comprised 20 games of 10 rounds each. Cooperation in rounds 9 and 10 clearly diminishes for several days, consistent with unravelling dynamics observed in prior work, but then appears to stabilize. (**b–d**) Cooperation as function of round (black lines) on days 1 (**b**), 11 (**c**) and 20 (**d**). Each day comprised 20 consecutive games of 10 rounds each, yielding 200 rounds in total. In all cases, coloured lines correspond to cooperation levels for rounds 1 (green), 8 (blue), 9 (purple) and 10 (red). The same pattern of unravelling in early days followed by stabilization is apparent.

higher than the usual range of 40–60% (refs 3,9,30,33). There are a number of reasons why our set-up may have led to overall higher-than-typical cooperation. First, although previous work[39,40] has found that players recruited from MTurk cooperate at similar rates to those in lab studies, it is possible that the recent evolution of the MTurk community has resulted in a population that is more cooperative than the usual, also non-representative[41], population of subjects present in traditional lab experiments. Second, prior work[10] has noted that cooperation rates in finitely repeated games are sensitive to choices in the game matrix parameters $g$ and $l$, where lower values correspond to more cooperation. As noted above, our values $g = 1$ and $l = 1$ were at the low end of previous studies, thus it is not surprising that we recover relatively high cooperation rates. Third, prior work[10] has also shown that the duration of a finitely repeated game is highly predictive of initial cooperation levels. Our games, which were ten rounds long, were relatively long compared with previous experiments; thus once again it is not surprising that cooperation levels were relatively high. Moreover, analogous logic would suggest that the overall duration of the experiment could also be related to cooperation levels. Because our design required us to inform participants about the length of the experiment, this knowledge may also have led to more cooperative behaviour. Finally, although players were not explicitly told the size of the population with whom they were being matched, they could have inferred this information from the counter in the virtual waiting room. Likewise, they were not directly informed that they were playing with the same population every day but could have inferred as much from their instructions, and hence could have reasonably concluded that they would anonymously encounter the same players several times over the course of the experiment. It is plausible, therefore, that the general expectation of repeated interactions also facilitated cooperative behaviour.

In other respects Fig. 1 shows that early behaviour closely resembled results from similar previous experiments. Specifically, Fig. 1b shows that cooperation levels, which remained high during the early rounds of each repeated game, dropped to a relatively low level in the final rounds, exhibiting the so-called 'end game' effect predicted by the rationality hypothesis[1]. Moreover, between games cooperation levels exhibited the well documented 'restart effect'[42] in which cooperation jumps sharply from the last round of game $j$ to the first round of game $j + 1$. Other than the relatively high average level of cooperation, therefore, the dynamics of session play was qualitatively similar to previous experiments of comparable duration[2,3,10,38]. Importantly, first session play also lends support to the rationality hypothesis: cooperation levels in round 1 (green line) increased slightly over the course of the session, but decreased steadily for rounds 9 (purple line) and 10 (red line), consistent with previous claims of unravelling[2,10,31]. Also importantly, Fig. 1a shows that the decrease in cooperation during rounds 9 and 10 continued for several days, but then slowed dramatically for the remainder of the experiment. Supporting this claim, Fig. 1c,d show that cooperation levels on days 11 and 20, respectively, continued to start high for each game and drop sharply as the end-game approached, but that there was much less change over the course of a session. Moreover, the relatively small decreases in rounds 9 and 10 cooperation that did occur over the course of a session largely 'reset' themselves at the start of the next session such that there was little change from day to day.

**Unravelling stabilizes after several days**. Figure 2 shows the same general trends in three different ways. First, Fig. 2a shows the average rate of cooperation by round, broken down by day. Consistent with the observations from Fig. 1, the pattern of

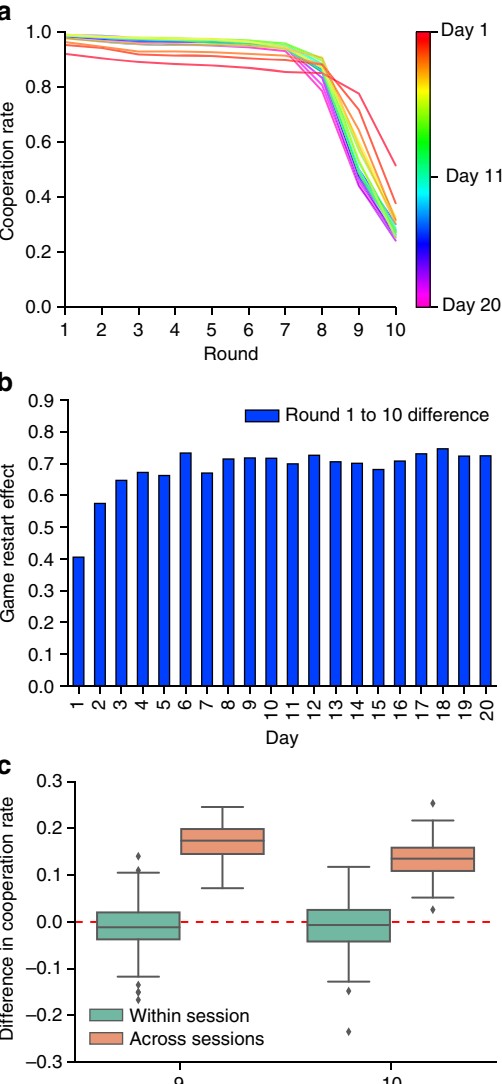

**Figure 2 | Stabilization of cooperation.** (**a**) Cooperation by round averaged over the course of a 20-game session, grouped by day. Early days (coloured red through green) show the sharpening of end-game effect (that is, initial cooperation increases but drops off further and more suddenly as the end-game approaches), after which the pattern stabilizes (green through purple). (**b**) Average restart effect between games (that is, difference in cooperation rate between round 10 of game $j$ and round 1 of game $j + 1$). Consistent with (**a**), the restart effect increases for several days then stabilizes. (**c**) The stabilization of cooperation is partly accounted for by the cross-session restart effect (orange): the jump in cooperation rate between the last game of day $d$ and the first game of day $d + 1$ for rounds 9 (left) and 10 (right). For comparison, the corresponding within-session effect (that is, difference in round 9/10 cooperation rate between successive games within a session) is also shown (teal).

cooperation at first changes from day to day, increasing in early rounds and decreasing in later rounds, but then appears to stabilize after several days (green through purple). Second, Fig. 2b shows the daily average of the game restart effect—that is, the difference between round 10 on game $j$ and round 1 on game $j + 1$—over the course of the experiment. Again consistent with the results above, the restart effect increases sharply for the first several days as the end-game effect visible in Fig. 2a becomes more pronounced, but again it stabilizes after several days.

Finally, Fig. 2c shows the session restart effect (as distinct from the game restart effect): the difference in cooperation levels for rounds 9 and 10, respectively, during game 1 of day $d+1$ compared with game 20 of day $d$ (orange box plots). For comparison, Fig. 2c also shows the corresponding difference between successive games within the same session (green box plots). Whereas the across-game difference is slightly negative within a session, the across-session effect is large and positive (on 17.2 and 13.6% for rounds 9 and 10 respectively), largely accounting for the 'reset' effect noted above in Fig. 1.

Taken together, Figs 1 and 2 suggest that play can be broken into two phases: an 'unravelling' phase during which players start defecting on progressively earlier rounds, and a 'stable' phase during which unravelling abates. Addressing this question more systematically, Fig. 3 shows the distribution of round of first defection, $r_d$ for each day of the experiment. To identify the onset of a stable phase, we apply a two-sample Kolmogorov–Smirnov (K–S) test to successive days, finding that day-to-day changes are significant up to day 7 but then insignificant thereafter (see 'Methods' section for details). In addition, the onset of a 'stable' state at roughly day 7 can be inferred in at least two other ways: first, by noting the change of slope in the cooperation rates for rounds 9 and 10 (Fig. 1a); and second, by observing the between-game 'restart effect', which rises for the first several days and then stabilizes, again around day 7 (see Fig. 2b). Although these measures are less precise than the K–S test applied to the distribution of round of first defection, they both yield similar results. We therefore identify day 7 as the end of the unravelling phase (although we note that the precise day on which stabilization occurs is relatively unimportant for our results) and hereafter treat the period spanning days 7–20 as the stable phase.

Figure 3 also reveals three additional trends of interest. First, during the unravelling phase the left-hand bar—comprising a small group of early defectors—largely disappears, consistent with the assertion[10] that players first converge on one of a number of 'threshold' strategies. That is, they cooperate conditionally until some predetermined round $r_i$ after which they defect unconditionally (one player continued to defect in all rounds throughout the experiment). Second, among initially cooperative players there is a drift toward earlier first defection, again consistent with the conjecture that rational players, having settled on a threshold strategy, begin to slowly unravel. Finally, however, Fig. 3 also provides some direct evidence for the existence of a significant minority of players who do not appear to follow the unravelling pattern. Specifically, we observe that fully cooperative games occurred at rates between 15 and 20% for the duration of the experiment. Since players were paired randomly, and a game where neither player defected requires both players to be conditional cooperators, then a frequency of 16% of games with no defection implies a 40% frequency of conditional cooperators.

**Identification of resilient cooperators**. Summarizing, Figs 1–3 suggest that, consistent with the rational cooperation hypothesis, a majority of players first converge onto one of a number of threshold rules, and then subsequently exhibit 'unravelling' as their thresholds creep earlier with experience. Strikingly, however, Figs 1–3 also suggest that a significant minority do not exhibit this pattern, but rather consistently behave like conditional cooperators. To test for these different player types more systematically, we exploit the roughly 3,720 observations per player to identify individual-level strategies as well as their evolution over time. Specifically, we estimate for each player $i$ a unique strategy $s_i(j)$ for each game $j$ from among eleven predefined strategies: ten 'threshold' strategies $T_x$ for each round $x = 1, \ldots, 10$, according to which a player conditionally cooperates up to round $x - 1$ and then defects unilaterally from round $x$, and CC for players who conditionally cooperate for the duration of the game (see 'Methods' section for details). Figure 4a shows inferred strategies for the 94 players who completed the experiment: each row of 400 cells represents a single player $i$, where each cell is coloured to indicate $i$'s inferred strategy for a

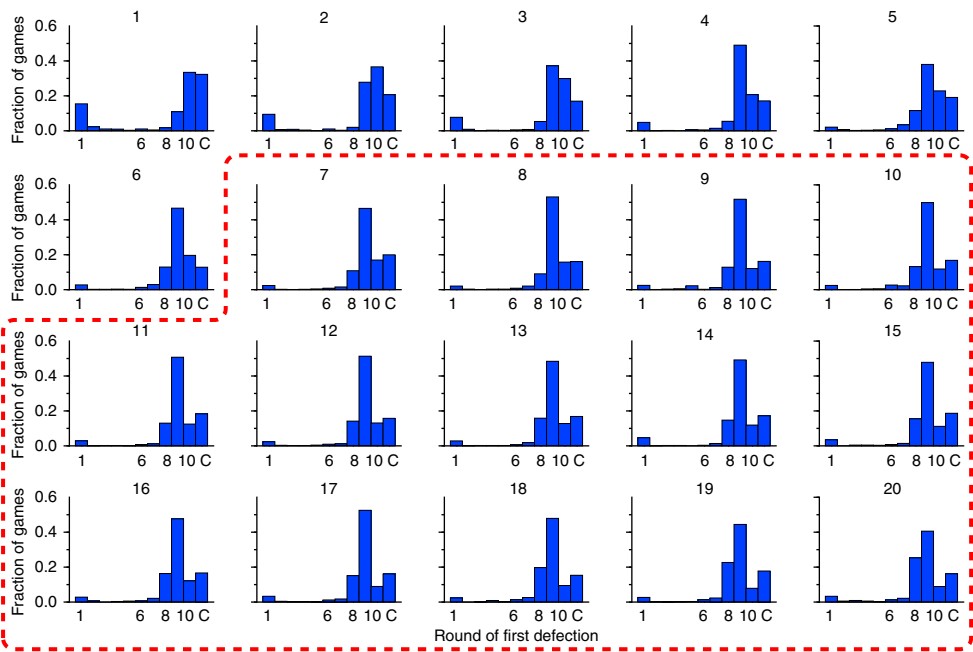

**Figure 3 | Stabilization of defection.** Distribution of round of first defection, $r_d$, over all games by day. The last bin, C, indicates games where neither player defected. In days 1–6, players appear to converge on one of a number of threshold strategies, in which they cooperate conditionally until some predetermined 'threshold' round $r_i$ and then defect unilaterally. During this interval the modal round of first defection also creeps earlier. The red highlighted region denotes the 'stable' phase of the experiment during which the distribution of round of first defection remains sufficiently similar from day to day that a K–S test is non-significant.

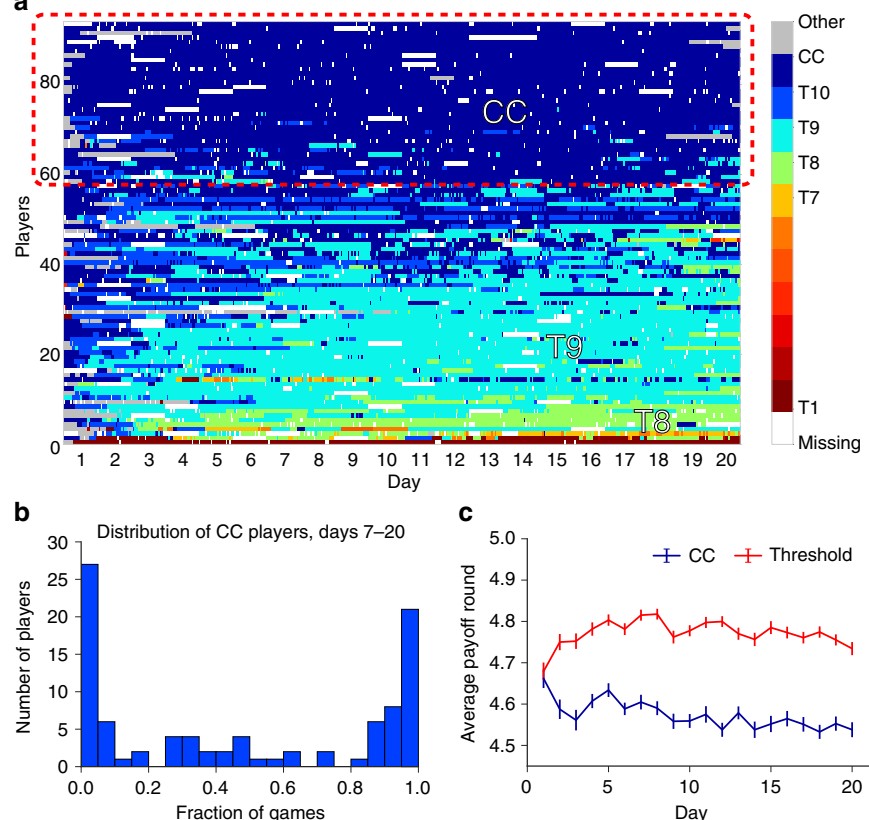

**Figure 4 | Identification of resilient cooperators.** (**a**) Inferred strategies over time. Each cell represents the strategy for a single player (row) for a single game (column), and is coloured by cooperativeness (more blue ⇒ more cooperative; more red ⇒ less cooperative). $T_1$–$T_{10}$ refer to 'threshold' strategies, where a player playing strategy $T_x$ conditionally cooperates up to round $x-1$ and then defects unilaterally from round $x$. $T_1$ corresponds to defection on every round and CC corresponds to full conditional cooperation (also known as 'grim trigger'). Grey regions refer to play that was not consistent with any of the assumed strategies. White regions refer to missing games. (**b**) Histogram of % of games classified as CC. The right-hand mode comprises 36 players who play CC in at least 80% of games; these players are identified as resilient cooperators. (**c**) Average per-round payoffs of players identified as resilient cooperators (blue line) and rational players (red line) respectively (averages are computed over games for each day; error bars are s.e.).

single game $j$. Figure 4a reveals three main results. First, consistent with previous work[2,3,10], the 11 predefined strategies account for a large fraction of all player-game observations; specifically, the fraction of 'other' strategies declines from about 19% on day 1 to <1% by day 7 (see Supplementary Fig. 6). Second, Fig. 4a shows that roughly 60% ($n=58$) of players exhibited behaviour consistent with the rational cooperation hypothesis: starting out playing CC but then switching to progressively less cooperative threshold strategies (that is, $T_{10}$, $T_9$, $T_8$, $T_7$). Third, however, almost 40% of players ($n=36$) displayed no such systematic unravelling tendency, consistently playing CC throughout the experiment. Figure 4b which shows a histogram of % games playing CC during the stable interval (days 7–20) shows that in fact these 36 players, who occupy the right-hand mode of the histogram, all play CC in at least 80% of games. Finally, Fig. 4c shows the average daily payoffs for the 36 players who played CC (blue line) versus that of the other players (red line): the two groups had similar payoffs on the first day, when all players were cooperating at similar rates; however, for all subsequent days CC players received lower payoffs than threshold players by a large and significant margin ($|t| > 5.3$, $P < 10^{-6}$ for each day $d \geq 2$).

On the basis of this evidence we conclude (a) that roughly 40% of players were 'resilient cooperators' who persistently behaved as conditional cooperators even at substantial cost to themselves; and (b) the remainder were 'rational' in that they cooperated only inasmuch as they believed it was in their selfish best interest to do

so. We also confirm this behavioural classification of resilient cooperators with self-reported evidence from an exit survey conducted at the completion of the experiment; of the 94 subjects who completed the entire experiment, 38 reported that they had intentionally cooperated as long as their partner did, and had resisted the temptation to defect first. Moreover, they reported that they had maintained this strategy throughout the experiment even after perceiving others to have behaved selfishly (see 'Methods' section for more details of self-reported strategies). Importantly we found that 33 of the individuals whom we identified as conditional cooperators in this manner were also among the 36 individuals in the right-hand mode of Fig. 4b, indicating extremely high agreement between quantitative and qualitative classification schemes (see Supplementary Fig. 7 for additional analysis of resilient cooperators by gender and age, and Supplementary Fig. 8 for analysis by experience).

**Resilient cooperators permanently stabilize cooperation.** The existence of resilient cooperators in turn suggests an explanation for the observed slowdown in unravelling: as the rational players learned the true fraction of conditional cooperators in the population, they converged on a 'partially unravelled' state that balanced the risk of exploitation by other rational players with the potential gains from cooperation with CC players. If correct, this explanation would also suggest that the observed slowdown was

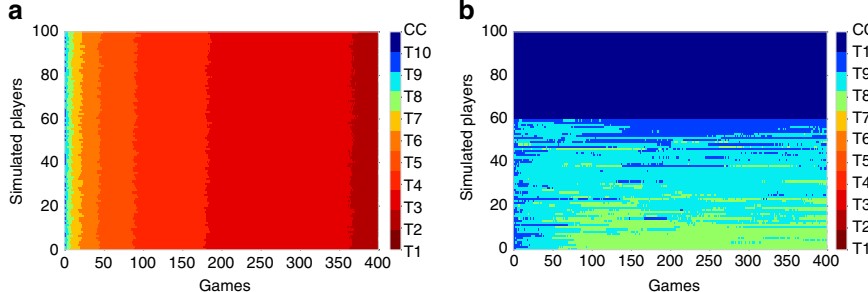

**Figure 5 | Resilient cooperators stabilize cooperation in an agent-based model.** In both cases, $T_1$–$T_{10}$ refer to threshold strategies, where a player playing strategy $T_x$ will cooperate conditionally up to round $x$ and then will defect unilaterally. $T_1$ corresponds to defection on every round and CC corresponds to full conditional cooperation (also known as 'grim trigger') (**a**) Individual strategies for 100 simulated agents over the course of 400 games in the absence of resilient cooperators (that is, all agents are rational cooperators who selfishly best-respond to the inferred distribution of strategies in the population). In this case cooperation unravels completely. (**b**) Individual strategies for the same model but with 40% resilient cooperators and 60% rational agents. In this case cooperation stabilizes after 100–150 games (equivalent to 5–8 days)

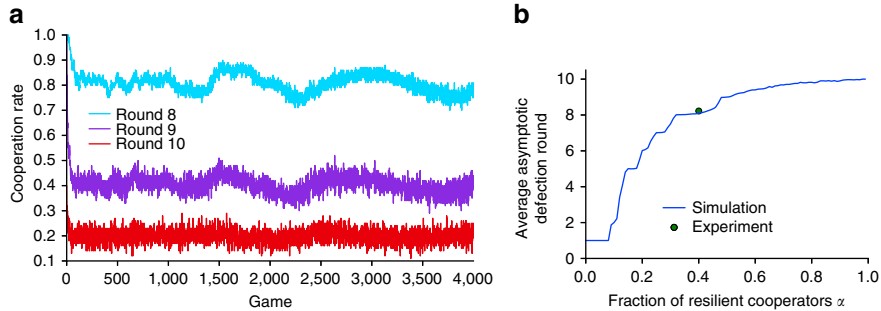

**Figure 6 | Asymptotic behaviour of the simulation model.** (**a**) Cooperation by game for rounds 8, 9 and 10 for the simulation with 40% resilient cooperators. The timescale is 4,000 games, ten times the length of our experiment, suggesting that the stabilization of unravelling observed in Fig. 5b is permanent. (**b**) Asymptotic average round of first defection $r_\infty$ versus fraction of resilient cooperators $\alpha$, averaged over 10 sets of simulations. The single point shows the values ($\alpha$, $r_\infty$) obtained from our experiment.

permanent and that cooperation levels by the end of the experiment were close to their asymptotic limit. To test these related hypotheses we simulated an agent-based model comprising two types of agents: resilient cooperators who unconditionally play CC for the entire duration; and 'rational' players who continually update their beliefs about the distribution of player types in the population and then choose among available threshold strategies $T_x$ so as to maximize their expected payoff given their beliefs. Specifically, in each game the rational players: (a) form beliefs about the strategies being played by other agents based on their past opponents' play; (b) conditional on these beliefs, calculate their expected utility for each available strategy; and (c) stochastically update their current strategy in proportion to each potential strategy's expected utility (see 'Methods' section for details). By systematically varying the fraction $\alpha$ of resilient cooperators we can explore their impact on unravelling.

Figures 5a,b show the results of the simulation for $\alpha = 0$ and $\alpha = 0.4$, respectively, for $N = 100$ agents. In the absence of resilient cooperators (Fig. 5a), rational players exhibit exactly the unravelling predicted by the rational cooperation hypothesis[1,2,10,31]: over the course of 400 games, players unravel almost uniformly through $T_{10}$ all the way down to $T_1$, albeit progressively more slowly for lower thresholds. In contrast, when 40% of players are resilient cooperators (Fig. 5b), corresponding to what we observed in our experiment, unravelling is curtailed, with $T_9$ emerging as the modal strategy and significant fractions occupying $T_{10}$ and $T_8$. Encouragingly Fig. 5b bears a close resemblance to Fig. 4a, suggesting that in fact the entire

distribution of steady-state strategies of agents in the simulation is similar to that for our experimental subjects.

In addition to replicating the high-level results of our experiment, the learning model also makes two predictions. First, as shown in Fig. 6a, cooperation in rounds 8, 9 and 10 for the $\alpha = 0.4$ case remains stable for at least 4,000 games, ten times the length of our experiment. This result suggests that the apparent stabilization of cooperation that we observe in the experiment after 7 days is not simply a slowing down of the unravelling process, but an end to it. In other words, the model predicts that with sufficiently many resilient cooperators present in a population of rational cooperators, cooperation can be sustained indefinitely. Second, the model also makes a prediction about how many resilient cooperators are necessary to sustain cooperation even among rational cooperators. To show this result, we first define $r_\infty$ as the average first round of defection $r_d$ for rational players as it approaches its asymptotic limit (in practice we estimate $r_\infty$ by running the simulations for at least 2,000 games). Figure 6b shows estimated $r_\infty$ as a function of $\alpha$ along with the values of $\alpha \approx 0.4$, $r_\infty \approx 8.2$ obtained from our experiment (averaged over the stable phase, days 7–20). In addition to reinforcing the agreement between experiment and simulation noted above, Fig. 6b also predicts the full functional dependency of $r_\infty(\alpha)$. Notably, $r_\infty$ appears to undergo a sharp transition, resembling an epidemic threshold[43], at some critical value $\alpha_* \approx 0.1$: for $\alpha < \alpha_*$ unravelling progresses all the way to the beginning of the game ($r_\infty = 1$), whereas for $\alpha > \alpha_*$, $r_\infty$ increases sharply and nonlinearly, eventually approaching $r_\infty = 10$ (that is,

no unravelling) when $\alpha = 1$ (see Supplementary Figs 9 and 10 for robustness checks).

## Discussion

Our experimental and simulation results support three conclusions. First, roughly 60% of the player population is 'rational' in the sense proposed by Kreps et al.[1] and thus is susceptible to 'unravelling' dynamics noted by previous experiments[10]. Second, however, roughly 40% of the player population is not rational in this sense, instead playing CC for the duration the experiment even as they are exploited by the rational majority. Finally, the existence of these resilient cooperators appears to stabilize the unravelling dynamics after several days, thereby conferring long-run benefits on both the resilient minority and the rational majority. Strikingly, the overall rate of cooperation stayed above 84% throughout the experiment, meaning that players collectively extracted roughly 84% of the maximum average payout possible. Our results therefore cast prospects for long-run cooperation in a hopeful light; as long as a sufficiently large minority of people are determined to act as conditional cooperators, high levels of cooperation can be sustained indefinitely even when the majority is willing to cooperate only when it is in their pragmatic self-interest to do so. Interestingly, this long-run cooperation appears to be stable even in the absence of reinforcement mechanisms such as punishment[44,45], reputation[7,9,30] and partner-selection[38,46].

We note that the observed fraction of resilient cooperators is reminiscent of previous findings that between 40 and 60% of players in social dilemmas choose to cooperate[4,8,20,28,29]. Whereas these results refer generally to one-shot games, however, we find that resilient cooperators continue to cooperate even in the face of persistent exploitation by rational players; thus our result builds upon and strengthens previous claims. For example, Fig. 4a shows that more than 40% of players initially appeared to play CC, only to exhibit unravelling after several days. Moreover, self-reports also indicate that some of these players began with altruistic motivations but succumbed to unravelling in the face of prolonged exposure to less altruistic players. As one player reported, 'I started off by trying to cooperate...but as time wore on it became evident that I was only being cheated over and over again in the final rounds... I started to see my daily bonus go down and inevitably began defecting first on the last round, then in the 9th, and finally in the 8th...' Based either on behaviour or on subjects' own self-evaluation, therefore, an experiment conducted for a single session would have substantially overestimated the number of conditional cooperators. Our finding also complements an earlier claim[24,47] that some individuals exhibit a 'cooperative phenotype' that is stable across different cooperative games. Whereas these claims refer to correlations between player contributions in one-shot games, however, we find that resilient cooperators retain a highly consistent strategy over many repetitions of the same game; thus the two claims refer to different kinds of inter-temporal consistency. In both cases, our results demonstrate that the ability of 'virtual lab' experiments to run for much longer timescales than is possible in traditional lab settings allows them to uncover empirical regularities that are both novel and also of theoretical interest.

Although our experiment demonstrates the existence and importance of resilient cooperators, it does not settle the question of what motivates some players to resist unravelling when so many others succumb to it. Examining responses to the exit survey, we suggest four possible motivations. First, a number of players cited the welfare of other players as a reason for cooperating (for example, 'I tried to get the best outcome for me and the other person'), and hence could reasonably be labelled 'altruistic' in the sense that they exhibit other-regarding preferences[3]. Second, several players invoked a desire to achieve 'fairness,' or expressed guilt at having defected first, both which are as consistent with norm-based accounts of cooperation[7,13,20]. Third, other players appear to have been motivated largely by self-interest, declaring that the long-run nature of the experiment rendered cooperation 'rational.' Finally, others still appear to have cooperated reflexively, giving no further reason (for example, 'I chose to cooperate unless the other person chose the defect'). Because our experiment was not designed to disambiguate between these four theoretically distinct explanations we refrain from ascribing any particular motive to resilient cooperators, leaving the matter to future investigation.

In addition to the question of motivation, our experiment also exhibited several other limitations that in turn raise questions for future work. First, as noted earlier the levels of cooperation observed in our experiment were high relative to previous experiments. Although we have proposed a number of possible explanations for this observation, a definitive explanation awaits further study. Second and relatedly, while interactions were anonymous and our population was large relative to traditional lab experiments, players could have reasonably inferred that they would play against the same players repeatedly; thus the long run sustainability of cooperation when players have no such expectation remains an open question. Third, the fraction of resilient cooperators $\alpha$ is likely a function of the parameters $l$, $g$ and $H$. Understanding this function, and how it interacts with the unravelling of rational players therefore remains an open question that could be addressed in future experiments. Fourth, although our simulation model did account for the extent of unravelling, it did not account for some of the strategies that we observed, nor was it able to replicate some features of our data (for example, the 'session restart effect'). More sophisticated and realistic models may therefore shed additional light on our empirical results. Finally, the simulation results motivate a specific additional hypothesis—that in order to be effective the number of resilient cooperators in a population must exceed a certain critical mass $\alpha_*$—that could be the subject of future experiments.

## Methods

**Recruiting.** To resolve the logistical challenges of obtaining a subject pool willing and able to commit to such a lengthy experiment, we recruited in advance a panel of several hundred subjects from Amazon's Mechanical Turk, a crowdsourcing site that is increasingly used by behavioural researchers for recruiting and paying subjects[40,48,49]. To maximize the likelihood that subjects would remain in the experiment for the full duration we took the following steps. First, we advertised the experiment roughly a week in advance of commencement, and requested that subjects only agree to participate if they were willing to participate for the entire duration of the experiment. Second, we asked volunteers to indicate at least three time intervals during which they expected to be available, and scheduled the two experiment sessions at times of day that were both popular and feasible (13:00 and 15:00 hours ET). Third, we divided the panel randomly among these two sessions, and notified participants that only those who made it on the first day (4 August 2015) would be able to participate for the entire experiment; this group comprised 113 participants. Finally, we warned subjects that they would be excluded from the experiment if they missed more than two sessions, and reminded them each evening that we were expecting them to return the following day. At the end of the experiment, 94 participants remained.

**Attrition analysis.** Although our retention rate (94 out of 113) was extremely high relative to previous experiments conducted using Amazon's Mechanical Turk[31], it is nevertheless possible that subjects who missed more than two sessions and were excluded from further participation (henceforth referred to as 'dropouts') were systematically different from those who completed the experiment, and hence that our results are biased by the selection method. We first compare the average cooperation rate of the 19 players who dropped out to the main 'completer' population. Since players' behaviour was not stationary over the course of the experiment, we compare each dropout to the main population over the range of days they participated. For example, suppose that a particular player dropped out on day 9. We compare the average per-game cooperation rate of this player with

the 94 completers for days 1 through 9. Supplementary Fig. 11 compares the cooperation of dropout players with the expected population distribution using a standardized $z$-score for each dropout. Of the 19 dropouts, only one player was in the most extreme 5% of this distribution; thus we cannot reject the null hypothesis that the two samples were drawn from the same underlying population. Second, we also compare players who dropped out with those who completed the experiment by estimating following linear models for $y_i$(day), denoting the average per-game cooperation rate for each player $i$ on any day of the experiment:

$$y(\text{day}) = \beta_0 + \beta_1 \text{day} \tag{1}$$

$$y(\text{day}) = \beta_0 + \beta_1 \text{day} + \beta_2 \text{dropout} + \beta_3 \text{dropout} \times \text{day} \tag{2}$$

$$y_i(\text{day}) = \beta_0 + \beta_1 \text{day} + \beta_2 \text{dropout} + \beta_3 \text{dropout} \times \text{day} + \beta_{4,i} + \beta_{5,i} \text{day} \tag{3}$$

The parameters $\beta_0$ and $\beta_1$ estimate the baseline level of cooperation and change over time, respectively, while $\beta_2$ and $\beta_3$ capture differences in these parameters for players who dropped out. $\beta_{4,i}$ denotes one parameter for each player representing individual differences in baseline cooperation rates, and $\beta_{5,i}$ captures how players may individually change over time. As Supplementary Table 1 shows, the dropout terms are not significant at the 5% level in any of these models, indicating that players who dropped out neither began with a significantly different level of cooperation nor significantly changed vis-a-vis the main population over time. Finally, we corroborate these quantitative results with a qualitative analysis of exit survey responses for players who dropped out: almost all such players mentioned that they missed sessions due to forgetfulness or circumstances outside of their control. Of the players who dropped out, only one mentioned that the payment in the study was insufficient, and no participants indicated that they stopped participating due to a change in behaviour of other players (see Supplementary Methods for survey questions). For all these reasons, we believe there are no significant differences between players who we excluded due to absences and the rest of the population, and conclude that our results for players' strategies are unaffected by the exclusion of some players.

**Identification of the steady state.** We use a two-sample K–S test to detect significant changes in the distribution of round of first defection (Fig. 3) on successive, adjacent days. A statistically significant difference between days implies that the distribution of first defection is changing, whereas a non-significant difference implies the distribution is changing by an amount that is too small for us to detect, if at all. Since partners are randomly assigned and anonymous we assume that different games in the same day are independent but we do not assume any independence between rounds of the same game. Although the K–S test is not perfectly suited to our case (it is designed for continuous distributions whereas ours is discrete), it is arguably better than the alternatives such as the Mann–Whitney (MW) $U$-test, which is designed for ordinal numbers. Nevertheless, as a tentative robustness check we have also conducted a MW test on the same data. Supplementary Table 2 shows that there are significant changes in behaviour (as measured by the round of first defection) after each of the first 6 days of the experiment, and that this change becomes insignificant thereafter. Although not identical, the MW test generates broadly similar results, thus we denote the 'steady state' of the experiment as days 7–20.

**Inferring strategies.** To infer the strategies that players are using from their observed play we assume that players are selecting according to the following possible strategies: 'threshold strategies' $T_x$ ($1 \leq x \leq 10$) according to which a player conditionally cooperates up to round $x - 1$ and then defects unilaterally from round $x$ (note $T_1$ corresponds to the ALLD (always defect)); and CC (also known as 'grim trigger') according to which a player cooperates until his or her partner defects, and then defects thereafter. We also observed a handful of players who played a variant of a $T_x$ strategy, where in the event that their partner cooperated on round $x$ they would switch back to cooperating. Because these players were very rare, and because their subsequent cooperation lasted at most for a round or two, for simplicity we have coded them as $T_x$. Finally we code as 'other' play that does not conform to any defined strategy (see Supplementary Fig. 6 for fraction of players classified as 'other').

For each player $i$ and each game $j = 1 \ldots 400$ we then define

$$1_{i,s}(j) = \begin{cases} 1 & \text{if player } i\text{s play in game } j \text{ is consistent with strategy } s \\ 0 & \text{otherwise} \end{cases}$$

Next we estimate the weight $w_{i,s}(j)$ placed on strategy $s$ for player $i$ as $w_{i,s}(j) = 1_{i,s}(j) + \gamma w_{i,s}(j-1)$ where $0 \leq \gamma \leq 1$ represents the discount rate for past behaviour (computing an exponentially weighted moving average). For each game $j$ we then assign to player $i$ a unique strategy $s_i^*(t)$ such that $w_{i,s^*}(t) = \max_s w_{i,s}(j)$. In the event of a tie between one of our eleven known strategies and an unidentified (?) strategy, we choose the known strategy. In the event of a tie between a $T_x$ strategy and a CC strategy, we choose the $T_x$ strategy. Figure 4 shows the output of this procedure: each row represents a single player $1 \leq i \leq 94$, each column is a single game $1 \leq j \leq 400$ and each cell is coloured according to the unique strategy

$s_i^*(t)$. Here we used $\gamma = 0.818$, approximating an exponentially weighted moving average with a 'period' of 10 games—half of a day in our experiment.

**Modelling player behaviour.** We study a 'smoothed fictitious play' model, which is a widely used learning model in repeated games[50]. The model is specified as follows. First, we assume that in each game, agents play one of a fixed number of strategies. For simplicity, we include only the threshold strategies, $T_x$, which account for the vast majority of observed human actions, including ALLD ($T_1$) and CC (see Fig. 4). Second, each agent $i$ maintains a vector of counts of the strategies his opponents played up to game $j$ denoted $\pi_i(j)$. Third, given their beliefs about others (that is, $\pi_i(j)$), before each game $j$ agents compute their expected utility $u_i(s) = \mathbb{P} \frac{\pi_i(j)}{\|\pi_i(j)\|}$, where $\mathbb{P}$ denotes the $11 \times 11$ matrix of payoffs to player $i$ when playing strategy $s$ and the opponent plays strategy $t$. Finally, agents choose their strategy $s$ for the next game with probability $\frac{\exp(u_i(s)/\beta)}{\sum_s \exp(u_i(s)/\beta)}$, where $\beta$ controls the amount of randomness in the decision ($\beta \to 0$ implies deterministic selection of the strategy with the highest expected utility, whereas $\beta \to \infty$ corresponds to uniformly random choices). The specific results reported in Fig. 6 used a particular value of $\beta = 0.005$. In Supplementary Fig. 10, however, we show equivalent results for a range of $\beta$ values ($0.001 \leq \beta \leq 0.1$) showing that the model's behaviour is extremely insensitive to the particular choice of $\beta$.

**Classifying exit survey responses.** As mentioned in the main text, 94 out of the original 113 subjects completed at least 18 out of 20 sessions. These 94 subjects were then asked to complete an exit survey for which they were compensated separately; all 94 subjects completed the survey (see Supplementary Methods for full text of the exit survey). Of particular interest to the current analysis are the responses of subjects to the following questions: (1) In choosing your actions in each game, what particular plan or strategy did you settle on, if any? Please describe it in your own words. (2) If you followed a strategy, did it change over the course of the study? What were your previous strategies? When and why did they change? Based on subject responses to these two questions, we coded a player as a resilient cooperator if they claimed they had always or almost always cooperated, and they claimed to have resisted the temptation to defect earlier even after witnessing others defect. Conversely we coded them as rational if they conceded that they had tried to benefit by defecting before others and they indicated that they had cooperated less over time. In addition, we also coded rational players specifically as $T_x$ if they mentioned a specific round on which they decided to unilaterally defect. Supplementary Table 3 shows excerpts of responses to the questions above, our coding of the responses, and also their inferred strategy from above.

**Data availability.** The data generated and analysed during the current study are available on the Open Science Framework at https://osf.io/64z8u/. The source code used to run the experiment is available at https://github.com/TurkServer/long-run-cooperation.

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

## Acknowledgements

We are grateful to Joseph Risi for analysing the exit survey responses, and to Jake Hofman and Winter Mason for statistical advice. We also thank the many Mechanical Turk workers who participated in this experiment.

## Author contributions

A.M. and L.D. built the experiment software, contributed to the experimental design, conducted the experiments, and analysed the data. S.S. and D.J.W. contributed to the experimental design and supervised the conducting of the experiments and the analysis of the data. A.M., S.S. and D.J.W. wrote the paper.

## Additional information

**Competing financial interests:** The authors declare no competing financial interests.

**Publisher's note**: 

