## [Peer Review File · Nature Communications]

Reviewers' comments:

Reviewer #1 (Remarks to the Author):

Authors show that altruists stabilize long-run cooperation in the finitely repeated prisoner's dilemma. Relatively long-run human experiments have been performed with the goal of testing the altruism and the rationality hypothesis, both of which are important in explaining cooperation in social dilemmas. During the early stages of the experiment the rationality hypothesis proved valid, but later on altruism also emerged, which ultimately stabilized cooperation.

How and why cooperation emerges in social dilemma games is an intensely investigated subject with obvious practical ramifications. The subject is popular in economics, the social sciences, and even in statistical physics and applied mathematics. Accordingly, recent research has shed light on the problem from many different perspectives, and also outlined many different ways and mechanisms that can stabilize cooperation. In this sense, the study addresses a relevant problem, with potentially far-reaching implications.

Unfortunately, the authors overlook an enormous amount of very closely related experimental and theoretical research on the subject, and it thus seems they know very little about the subject they are trying to advance. I can of course only speculate whether this negligence of preceding research is intentional or not, but either way, as it stands, because of this alone, I would consider the manuscript unpublishable, let alone in a prestigious journal like Nature Communications.

Task one for the authors is to catch up properly with the literature on this subject. Below is a list of papers, all dealing experimentally (by means of human/economic experiments) with cooperation in the prisoner's dilemma game or closely related social dilemma games.

Scientific Reports 5, 7843 (2015)
Scientific Reports 5, 10282 (2015)
Scientific Reports 4, 6458 (2014)
Nature Communications 5, 4362 (2014)
Scientific Reports 4, 4615 (2014)
Proceedings of the National Academy of the USA 109, 12922-12926 (2012)
Dynamic Social Networks Promote Cooperation in Experiments With Humans (2011) PNAS
Positive Interactions Promote Public Cooperation (2009) Science
Cooperating with the Future (2014) Nature
Static Network Structure Can Stabilize Human Cooperation (2014) PNAS

This is just works from two groups that I am aware of. Surely there is much more, which the authors will discover if they look up citations to the above papers.

In terms of theory, the iterated prisoner's dilemma game has been studied literally to death by various fractions of the aforementioned contingents of social and natural sciences. Perhaps the most recent notable advances are related to games on networks, as reviewed in Evolutionary games on graphs, Physics reports 446 (4), 97-216 and Coevolutionary games - A mini review, BioSystems 99, 109-125, for example.

The introduction needs a major rewrite and update for comprehensiveness and coverage of the field. The reported findings need to be placed in the proper context, compared, and discussed in the light of what so many other have done on this subject before.

Coming to the research itself, I have a very hard time buying the arguments of the authors that have to do with the duration of this experiment, and in particular, how altruism suddenly emerges. If anything, if cooperation suffers initially due to rationality, this will only make things worse and discourage cooperation even among those that perhaps might have considered it viable at the

start. The reported findings do not add up at all if compared to existing results on the subject. This is something that also needs to be carefully addressed and discussed, because it is completely tangential to expectation and reason.

Lastly, I find the color maps very crowded and unclear. There should be a better way to present the results graphically, and if vying for publication in top journal, the authors really should do better technically when communicating their results.

Reviewer #2 (Remarks to the Author):

This manuscript presents result of an experiment involving about 100 subjects playing a very long series of iterated PDs, about 400, along consecutive weekdays of a month. This is a very novel setup and the results are certainly relevant and provocative, and I am sure that the community of researchers in the wider field of behavioral sciences will be interested in them. Having said that, I do believe also that the manuscript would benefit from a number of improvements on the results presentation and that more details are needed to allow the reader to appreciate and understand what the authors are doing. Therefore, I recommend that the manuscript is returned to its authors so they can address the comments and suggestions below which, in my opinion, should be fully clarified and considered for the manuscript to be publishable with claims established on firm grounds.

Major remarks

1. In the abstract, and in a few other places in the text, the authors claim that players that display consistent altruistic behavior do so because they "resist[ing] the temptation to unravel out of a sense of collective welfare". While this may well be the case, the authors provide no evidence supporting this interpretation on solid grounds; it is perfectly possible that altruistic subjects are "programmed" to play like that without no consideration of collective welfare whatsoever. Other alternative explanations are also possible. In my opinion, if the authors want to make this interpretation of the observation of altruistic behavior, they need to support it much more clearly, and they can do so in principle by looking at their extensive questionnaire answers. Otherwise, they should either make no interpretation or discuss alternative interpretations.

2. Regarding the discussion of the experimental setup, reading the summary of page 2 one question arises naturally, namely, did subjects know how many of them were around? In other words, given that most players played 400 games and there are about 100 subjects, on average they played four times against each other subject. In fact, this number can be more than twice as large as subjects play on two different time intervals, so the pool was half the size. Can the participants infer this from the info available to them? If that were the case, there is the possibility that an implicit reciprocity is playing a role in the results, and it should be discussed properly. Otherwise, were they lied to and told that they would always meet new partners, or the re-pairing was phrased in a neutral manner? These are important issues to understand the details of the experiment and should be clearly stated and discussed.

3. While I am aware that there are some studies (e.g., Refs. 23 and 24 in the manuscript) that show that using Amazon Mechanical Turk is OK for experiments and gives similar results to the lab, others (sharing some authors) show large discrepancies [e.g, Rand et al., PNAS 111, 17093 (2014)]. Therefore, it is not clear to me whether this particular experiment would yield the same results in a lab. In particular, I see four possible problems here. First, given that the number of subjects is not exceedingly large, if some of them are playing under different names in the Turk this could induce a huge bias in the results. Can the authors check that this is not the case? Second, due to the fact that the experiment carries over during a month, it is perfectly possible that the subjects looked for info on the game they were playing, and therefore adapted their

behavior accordingly. Can the authors be sure that the subjects remained "naive" during the experiment? Which consequences would the search for information have on the results? Third, I assume that of course the Turk identities involved in the experiment are the same all along, but can the authors be sure that the subjects are the same? Are there any tests or checks that aim to ensuring that the person is the same and is not leaving his/her account to somebody else? In a typical experiment one would not expect that this would be an issue, but in this long setup, it is perfectly possible that if somebody can't join one of the days, allows somebody else to play on his/her name. Finally, in the questionnaire players are asked whether they have participated before in similar games on the Turk. Was any action taken if the answer to this question was affirmative, i.e., the subject was excluded? Note that this is connected to my remark above about having information on the game or being "naive".

4. Still about the issue of the experimental setup, unless I overlooked something there is a lot of information missing about the setup. How are subjects paid? Do they get a exchange rate of the total number of points, are some games selected at random,...? All we know from the text (again, unless I'm wrong and if that's the case I'm sorry, but perhaps it needs to be more salient) is that they receive \$20 if they complete 18 sessions, as noted in the Methods section, but nothing else is said about the payment scheme. Another thing we do not know is the average payment the subjects get and, in connection with this, where are the subjects from. This is important because Turkers may be from many different countries and the payment should be appropriate to the country chosen and, if there are players from different countries, it is possible that payments are large for some and small for other. From the questionnaire (see p 10 of SI) it seems participants come from the US and Canada but it should be stated in the text and also how many come from each country, if this is indeed the case. Finally, the authors should include the full instructions (as is customary in experimental economics) as shown to the participants so readers can check exactly what the subjects knew and how it was presented to them.

5. Going now into the results, I find it remarkable the high level of cooperation one can see in Fig. 1, particularly in panel F. It seems very large compared to the studies reported in the meta-analysis by Embrey et al. (Ref. 4 of the manuscript). If one looks at Table 1 in that paper, for values of g and l closer to the ones used here ($g=l=1$) one sees that the level of cooperation reported in the study AM1993 is considerably smaller, even in the last super game. What can be the reason for this? This is important because it appears (admittedly, maybe by chance) that there is a kind of trend in the results by some of the authors of this manuscript. For instance, looking at the paper by Wang, Suri and Watts □(Ref. 17 of the manuscript) and comparing the results to similar papers by Gallo and Yan [PNAS 112, 3647 (2015)] and by Cuesta et al. [Sci. Rep. 5, 7843 (2015)] one sees that the cooperation levels are much higher in the former. Parenthetical remark: the authors might consider quoting these papers when mentioning reputation at the bottom of page 12, as they show that the cooperation observed on Ref. 17 arises from reputation.

6. While I very much like the classification of strategies (but see below under "Minor remarks") and the agent-based model designed on the basis of that classification, I believe that a more quantitative comparison of the results would be in order. For instance, on panel D in Fig. 4 (note the typo "resilent" in the horizontal axis label) we are shown a point corresponding to the experiment that is practically on top of the simulated line. However, if I understand correctly the line is an average, so there should be a standard deviation interval around that line; error bars or shaded regions are needed in that plot. In addition, I believe that the comparison could involve the distribution of threshold strategies in the experiment and in the simulations: if these distributions were really similar that would be a very nice point for the simulation model.

7. In the conclusions, on page 11, the authors state that their experiment supports the conclusion that roughly a 40% of the population behave altruistically. This is something that, provided all the points mentioned in this report are satisfactorily sorted out, I am ready to admit. In fact, I believe that the authors should comment on the "universality" of this result, namely that a fraction of people between 30% and 50% usually cooperate in the first round of prisoner's dilemmas or public

goods games (see e.g. the review by Ledyard on the Handbook of Experimental Economics by Kagel and Roth (Princeton, 1995), and even on static networks, see the metaanalysis in Grujic et al. [Sci. Rep. 4, 4615 (2014)]. In connection with this, it is interesting to mention that recently Peysakhovich et al. [Nat. Commun. 5, 4939] and Capraro et al. [Sci. Rep. 4, 6790 (2014)] have found that comparable fractions of the population are consistently cooperative across games and across cost-benefit ratios, and have even coined the term "cooperative phenotype" to describe these people. I believe that the authors should comment on this universal fraction of cooperators and on the possibility that these subjects are just born cooperators. Interestingly, a paper in the latest (at the time of writing this report) issue of PNAS by Yamagishi et al. [PNAS, Early Edition, May 2, 2016] seemingly connects altruistic giving in the DG with the thickness of the dorsolateral prefrontal cortex in the brain, which would point in the direction of some people being "natural cooperators" (cf. also my comment no. 1 above, this would be another alternative interpretation of the results).

8. As a final suggestion, and given the demographics reported on Fig. S2, I believe that the authors should check how the cooperator percentage is distributed among male and female participants, and also among different age ranges (in this last case I believe that they could just try to split the age range in three intervals with approximately the same number of subjects, otherwise the statistics may be poor). Whether the results show differences in terms of gender or age or not, it is an interesting result in itself (the correlation between these variables and the threshold strategies would also be very interesting). Differences in cooperative behavior across genders are a controversial issue in the literature, with reports in favor and against, and age is also becoming of interest in recent years, so this could be an interesting contribution from this study without much effort. Of course, the caveat about the reliability of these data on the Turk remains, but if the authors trust their subject pool, I believe these analyses should be done.

Minor remarks

1. On line 2 of the abstract, it is stated that "the evolution of cooperation in repeated games of prisoner's dilemma remains unresolved". This is the case only with finitely repeated games, there is no mystery in dyadic, infinitely repeated games, where infinite equilibria are possible and experiments show abundantly that cooperation emerges through reciprocity. I am sure that the authors are well aware of this, they probably forgot to add "finitely", but they should add it in a revised version.

2. Regarding the location of the transition to a stationary behavior, the authors report that using a Kolmogorov Smirnov test to compare the distributions shows that they become effectively indistinguishable from day 7 on. Have the authors tried other tests applied to the distributions? And have they tried other criteria based on other magnitudes, such as, e.g., cooperation per round in the different days? Would any of these lead to different results?

3. On page 7, line 2 from bottom, the authors report that the ten threshold strategies plus the CC strategy account for "the vast majority of observed behavior". This should be properly quantified. They later indicate on p. 8 that roughly 60% play threshold strategies and roughly 40% play CC. Why are not the exact numbers given? And, more importantly, do the authors have any idea of what the remaining players do?

4. The abstract is still formatted "Nature-style", as it appears that this manuscript has been transferred from Nature. This should be fixed and, given that the space restrictions are much more loose on Nature Communications, the authors should provide a good abstract, a proper introduction with a suitable revision of the literature, and all the necessary explanations and discussions of their setup and claims.

Reviewer #3 (Remarks to the Author):

The authors report results from an online experiment on the finitely repeated prisoner's dilemma. The experiment was run for 20 days using Amazon Turk, holding the subject pool constant. On each day, subjects played multiple instances of the 10-round prisoner's dilemma (against changing opponents). As the length of a game is known, theory would predict that subjects should learn over time that they should not cooperate in the last round, after which they should learn not to cooperate in the second to last round, up to a point when there is no cooperation at all. However, according to the experiment, such an unraveling does not occur - the vast majority of subjects cooperates at least up to round 8. The authors argue that these high cooperation rates are due to the presence of "altruists", who would cooperate even in the very last round, provided their co-player cooperated in all previous rounds. Using individual-based simulations, the authors show that if roughly 40% of the players are altruists (which is the value suggested by the experiment), then the behavior of the remaining 60% can be explained as a rational response.

The setup of the experiment is impressive. To the best of my knowledge, the authors are first to provide data of play in the repeated prisoner's dilemma over multiple days. This kind of data is not only interesting from an experimental perspective - it should also be extremely useful to theorists who want to study human learning in social dilemmas.

The presented results are remarkable, and the research in general seems to be well executed. Somewhat unfortunately, however, the authors failed to provide some information that seems to be rather essential to fully judge the quality of the paper (concerning details of the experimental design and of the statistical methods).

Provided these issues can be resolved, the manuscript certainly justifies publication in Nature Communications.

Major comments:

Quite essential information was missing. For example, I couldn't find information on the exact payoff values used in each one-shot game (i.e., the values of T , R , P , S); only the two derived quantities g and l have been reported on page 2. Without knowing these quantities, it is actually impossible to judge whether the reported results make sense, and whether they can be expected to be robust. Similarly, I would like to ask the authors to provide more information on

(i) how much time the experiment took (i.e., how many hours did it take the subjects on average to participate in this experiment)

(ii) average earnings per participant over the whole experiment (differentiating between fixed compensations and variable compensations)

(iii) average experience of the participants with previous social dilemma experiments

Similarly, I could not find information on how exactly the statistical tests were performed. In particular, the authors need to explain which statistical models they have used, and how they have taken into account that the decisions of different individuals cannot be taken as fully independent, and that also different decisions of the same individual cannot be taken as independent.

I was somewhat surprised about the relatively high cooperation rates in this experiment. Already in the very first round of the first game on day 1, subjects seem to cooperate with almost 90% probability. In the laboratory experiments on the prisoner's dilemma I know of, initial cooperation rates are typically much lower (see e.g. Ref. 3, Ref. 14, Hilbe, Röhl, Milinski, Nature Communications 2013; Xu, Zhou, Lien, Zheng & Wang, Nature Communications 2016).

I would like the authors to comment on this issue - are the high cooperation rates a consequence of the experimental design, of the chosen payoff values, or of the fact that Amazon Turk was used?

When classifying the players' behaviors, the authors only allow for strategies that have the property that if the strategy prescribes to defect in round r , the strategy also prescribes to defect in all subsequent rounds. How often did the authors observe behavior that was inconsistent with

this property (i.e., games in which a player defected in one round but cooperated in the next).

Papers in Nature Communications should have a formal Introduction and a Discussion section. Given the rather multidisciplinary readership of Nature Communications, the authors should make use of the Introduction section to explain in more detail what the "altruism hypothesis" and what the "rationality hypothesis" is (I am afraid many readers will not know the key references 1-4). Also, given the multidisciplinary scope of NatComms, it may be a good idea to have a somewhat broader bibliography that also covers results from biology, mathematics and psychology (at the moment, most of the articles cited have an economics background).

Minor comments:

(-) The variable g is used for two different purposes, to denote a particular payoff quantity (on page 2), and to refer to an instance of a game (e.g. on page 3).

(-) Coming from the evolutionary game theory literature, I find the term "altruists" somewhat unfortunate - most researchers in my field will associate altruists as people who cooperate in every single round, irrespective of the previous history of play. Thus I would recommend to use "conditional cooperator" or "grim trigger" instead, or at least to clarify the intended meaning of the word "altruist".

(-) Page 7: It is not immediately clear why the fact that between 15 and 20% of the games ended with full cooperation implies that 40% of the subjects would always cooperate until the co-player defected. Please explain in more detail.

(-) In the abstract and on page 11, the authors say that "the presence of altruists is both necessary and sufficient for cooperation to sustain itself" - I find this formulation somewhat inappropriate, as it pretends mathematical accuracy. Also, it seems to me that while altruists certainly help to sustain cooperation, the CC strategy described in the main text is not the only way how one could uphold cooperation.

(-) Figure S6: what would happen for larger beta values, e.g. $\beta=0.5$ or $\beta=1$? At the moment, the authors are entirely focusing on the case of "strong selection" (where players would most often adopt the best strategy), whereas researchers in evolutionary game theory are sometimes also interested in the case of "weak selection".

Reviewer #4 (Remarks to the Author):

Report for "Altruists stabilize long-run cooperation in the finitely repeated Prisoner's Dilemma" by Andrew Mao, Lili Dworkin, Siddharth Suri and Duncan J. Watts

Summary:

This paper presents the results of an experiment that studies cooperative behavior in a finitely repeated prisoner's dilemma (PD). Subjects, recruited via Amazon's Mechanical Turk, play a series of 10 round finitely repeated PDs over 20 sessions (each held at a different day). In each session, subjects play 20 separate finitely repeated PDs. Subjects remain anonymous throughout the experiment and are randomly assigned to new partners between games.

The experimental design allows the authors to study long-term behavior in this context. Contrary to standard theoretical predictions, main results show partial cooperation to stabilize after a limited period of unravelling (first 7 sessions). The analysis suggests a significant portion of the population to follow a conditionally cooperative strategy that never preempts defection. Simulation results resulting from a learning model (where people update their beliefs about the distribution of strategies used by the population) show that the presence of such subjects can rationalize the data and explain the partial unravelling in cooperation.

Comments:

The paper's main contribution to the literature is to directly study long term dynamics. Since subjects participate in 20 sessions of 20 finitely repeated PDs, there is direct experimental evidence of how subjects behave in such an environment after $20 \times 20 = 400$ distinct individual experiences with the game. The results are clear as further unravelling of cooperating looks convincingly unlikely in this context. However, I have several concerns with the interpretation of the results.

Most importantly, controlling for emergence of social norms and community enforcement (as in Kandori 1992) is an issue here given that a fixed number of subjects are repeatedly matched with each other over the course of a long experiment. Note that there are 94 subjects who are randomly matched with each other for 400 finitely repeated games. This implies that on average any two subjects interact roughly about 4 times. This can create dynamic collective reinforcement. I think there is some evidence that at least some subjects recognize this effect and choose their strategy accordingly. First, Figure 1 suggests Round 8 cooperation to decline slightly in the last few sessions of the experiment. Second, subject questionnaire responses at the end of the experiment indicate this type of thinking. I show some below:

#5: "I figured I might as well try to get others to adopt a better strategy. . . and more people started going with 5 and 5 all the way through."

#42: "I felt going beyond this was idiotic because in the end to continue in this fashion you are jeopardizing the whole groups pay."

#43: "Knowing that we were playing the same participants every day, I tried to learn what patterns others were playing so I could adjust my play to benefit me but still be fair."

#45: "It stayed mostly the same, I cooperated more than I thought I would. I guess I kind of hoped it would encourage others to continue to cooperate more as well."

#54: "I started out by trying to be cooperative. . . I was concerned that the more I defected, the more mistrust would seep into the game and the worse everyone would do (though my concern was with my own results, not others'). . . I was willing to lose a few pennies each game if it meant people cooperated for 8 or 9 rounds at least..."

I think it's important for the paper to address this concern. In several sections of the paper, it is stated that long term behavior strikingly appears to be stable in the absence of reinforcement mechanisms such as reputation or punishment. However, such reinforcement mechanism might be at work in this dynamic context. The key question is how much more unravelling would we expect to observe if subjects knew they would never interact with the same person, or further they would never interact with someone who interacts with this person in the future?

Session restart effects are very dramatic. Can the authors provide more insight on this? Could subjects be treating every session to be independent? The subjects might mistakenly believe that they are playing against a new group of people. Or it would be sufficient for them to believe that a significant portion of the subject pool makes such a mistake. Then the learning effects cannot carry through across different sessions. (There is also a question about how the learning model accounts for the restart effects that was not clear in the paper.) How do we interpret behavior in the last session in light of the restart effects? Is it long run as in after 400 repetitions of the game, or long run as in just 20 repetitions of the game. An interesting exercise would be to repeat a portion of this experiment where subjects play, for example, 80 finitely repeated PDs in one session. Play in the last repeated game here can be compared to play at the end of the 4th session in the original experiment. This should be indicative of to what extent results of this paper can be interpreted as "long-term" behavior.

Response to Referees

Below we have responded in detail to each comment. We have also made extensive changes to our manuscript. In particular we have made the following major changes:

1. Shortened Abstract
2. Added full introduction that situates our contribution more clearly both with respect to the specific literature on learning, which is largely in economics, and also evolutionary game theory.
3. Added more citations to the cooperation literature, including economics, evolutionary biology, complex systems research, psychology, sociology, and political science.
4. Added more details about the experimental design, subject population, player instructions, compensation, etc.
5. Added to discussion section:
 - a. A paragraph discussing similarities and differences with previous findings about cooperative behavior
 - b. A paragraph that raises four possible interpretations of observed behavior: “altruism”, “norms,” “long-run self-interest” and “reflexive.”
6. In light of 4(b) changed labeling of cooperative players from “altruist” to “resilient cooperators,” including in title
7. Simplified some figures and added new figures

We believe that the revised version addresses many, if not all, of the reviewers concerns and represents a very significant improvement over the original. We are grateful for the reviewers’ extensive feedback on the initial version and hope the paper is now suitable for publication.

Reviewer #1 (Remarks to the Author):

Authors show that altruists stabilize long-run cooperation in the finitely repeated prisoner's dilemma. Relatively long-run human experiments have been performed with the goal of testing the altruism and the rationality hypothesis, both of which are important in explaining cooperation in social dilemmas. During the early stages of the experiment the rationality hypothesis proved valid, but later on altruism also emerged, which ultimately stabilized cooperation.

How and why cooperation emerges in social dilemma games is an intensely investigated subject with obvious practical ramifications. The subject is popular in economics, the social sciences, and even in statistical physics and applied mathematics. Accordingly, recent research has shed light on the problem from many different perspectives, and also outlined many different ways and mechanisms that can stabilize cooperation. In this sense, the study addresses a relevant problem, with potentially far-reaching implications.

Unfortunately, the authors overlook an enormous amount of very closely related experimental and theoretical research on the subject, and it thus seems they know very little about the subject they are trying to advance. I can of course only speculate whether this negligence of preceding research is intentional or not, but either way, as it stands, because of this alone, I would consider the manuscript unpublishable, let alone in a prestigious journal like Nature Communications.

Task one for the authors is to catch up properly with the literature on this subject. Below is a list of papers, all dealing experimentally (by means of human/economic experiments) with cooperation in the prisoner's dilemma game or closely related social dilemma games.

Scientific Reports 5, 7843 (2015)

Scientific Reports 5, 10282 (2015)

Scientific Reports 4, 6458 (2014)

Nature Communications 5, 4362 (2014)

Scientific Reports 4, 4615 (2014)

Proceedings of the National Academy of the USA 109, 12922-12926 (2012)

Dynamic Social Networks Promote Cooperation in Experiments With Humans (2011) PNAS

Positive Interactions Promote Public Cooperation (2009) Science

Cooperating with the Future (2014) Nature

Static Network Structure Can Stabilize Human Cooperation (2014) PNAS

This is just works from two groups that I am aware of. Surely there is much more, which the authors will discover if they look up citations to the above papers.

We thank the reviewer for pointing out these citations. We agree that there is an enormous literature on the general topic of cooperation among humans that is spread across several disciplines, including complex systems and evolutionary biology—the focus of the above references—but also stretching back several decades in economics, sociology, political science, and psychology. We wish to reassure the reviewer that we are neither newcomers to the field (most of the papers mentioned above cite at least one of our previous papers; see [1-5]), nor were we strategically neglecting to cite particular research groups. Rather, our initial submission had originally been formatted for *Nature* and so was necessarily short. Given this constraint we focused our bibliography on prior work that was of direct relevance to our research question—long run learning in finitely repeated PD—which happens to be mostly in the economics literature. In light of the more flexible formatting requirements for *Nature Communications*, we have added a longer introduction that cites a much broader cross-section of literature, including some of the papers highlighted by the reviewer.

In terms of theory, the iterated prisoner's dilemma game has been studied literally to death by various fractions of the aforementioned contingents of social and natural sciences. Perhaps the most recent notable advances are related to games on networks, as reviewed in *Evolutionary games on graphs*, *Physics reports* 446 (4), 97-216 and *Co-evolutionary games - A mini review*, *BioSystems* 99, 109-125, for example.

These papers summarize the extensive literature on evolutionary game theoretic models of cooperation, the main objective of which is to account for how cooperative behaviors might have emerged among presumptively selfish actors over the course of evolutionary history. We are aware of this literature and have even contributed to it (see [2], cited by Perc and Szolnoki), however, it is

not directly relevant to the question of within-individual learning effects over the course of repeated play. In the Introduction we now clarify the distinction between learning effects of the sort we are studying and adaptation of strategies under selection pressure.

The introduction needs a major rewrite and update for comprehensiveness and coverage of the field. The reported findings need to be placed in the proper context, compared, and discussed in the light of what so many others have done on this subject before.

As requested, we have added a new introduction that cites more of the general literature.

Coming to the research itself, I have a very hard time buying the arguments of the authors that have to do with the duration of this experiment, and in particular, how altruism suddenly emerges. If anything, if cooperation suffers initially due to rationality, this will only make things worse and discourage cooperation even among those that perhaps might have considered it viable at the start. The reported findings do not add up at all if compared to existing results on the subject. This is something that also needs to be carefully addressed and discussed, because it is completely tangential to expectation and reason.

To clarify, we do not claim that altruism suddenly emerges. Rather we claim that 40% of the population are what we now call “resilient cooperators”, who conditionally cooperate from the start and maintain this behavior even in the face of unraveling. In other words, up to a point we find exactly what the reviewer suspects—namely that the initial “suffering” of cooperation due to rationality *does* make things worse and *does* “discourage cooperation even among those that perhaps might have considered it viable at the start.” That is precisely the nature of the unraveling that we observed for the first several days. What is new—and the reason why the timescale is important—is that the unraveling is not experienced uniformly across the population; rather about 60% exhibit signs of unraveling while 40% resist it. What we show is that the resilience of the 40% causes the unraveling to stop, thus benefitting everyone in the long run.

Lastly, I find the color maps very crowded and unclear. There should be a better way to present the results graphically, and if vying for publication in top journal, the authors really should do better technically when communicating their results.

In response to this concern we have split some of the more complex figures into separate figures, thereby increasing the overall number of figures but reducing their complexity. We have also added more explanatory text, thereby hopefully communicating the results more clearly.

Reviewer #2 (Remarks to the Author):

This manuscript presents results of an experiment involving about 100 subjects playing a very long series of iterated PDs, about 400, along consecutive weekdays of a month. This is a very novel setup and the results are certainly relevant and provocative, and I am sure that the community of

researchers in the wider field of behavioral sciences will be interested in them. Having said that, I do believe also that the manuscript would benefit from a number of improvements on the results presentation and that more details are needed to allow the reader to appreciate and understand what the authors are doing. Therefore, I recommend that the manuscript is returned to its authors so they can address the comments and suggestions below which, in my opinion, should be fully clarified and considered for the manuscript to be publishable with claims established on firm grounds.

Major remarks

1. In the abstract, and in a few other places in the text, the authors claim that players that display consistent altruistic behavior do so because they "resist[ing] the temptation to unravel out of a sense of collective welfare". While this may well be the case, the authors provide no evidence supporting this interpretation on solid grounds; it is perfectly possible that altruistic subjects are "programmed" to play like that without no consideration of collective welfare whatsoever. Other alternative explanations are also possible. In my opinion, if the authors want to make this interpretation of the observation of altruistic behavior, they need to support it much more clearly, and they can do so in principle by looking at their extensive questionnaire answers. Otherwise, they should either make no interpretation or discuss alternative interpretations.

This is a good point. Re-reading the exit surveys it does appear that at least some players considered collective welfare when playing CC; however, we agree that it is far from unanimous. In response to this observation, we have made two changes to the paper. First, we now refer to players who persistently play CC as "resilient cooperators" a label that we believe is justified purely on behavioral grounds. Second, we have added a paragraph to the discussion section in which we raise four potential interpretations of this behavior, including altruism, but also "programming," adherence to norms (as suggested by reviewer 4), and long-run self-interest. Because our experiment was not designed to disambiguate between these alternative hypotheses, we leave the task to future work.

2. Regarding the discussion of the experimental setup, reading the summary of page 2 one question arises naturally, namely, did subjects know how many of them were around? In other words, given that most players played 400 games and there are about 100 subjects, on average they played four times against each other subject. In fact, this number can be more than twice as large as subjects play on two different time intervals, so the pool was half the size. Can the participants infer this from the info available to them? If that were the case, there is the possibility that an implicit reciprocity is playing a role in the results, and it should be discussed properly. Otherwise, were they lied to and told that they would always meet new partners, or the re-pairing was phrased in a neutral manner? These are important issues to understand the details of the experiment and should be clearly stated and discussed.

Also a good point. Although the players were not explicitly told the size of the population with whom they were being matched, nor were they directly informed that they were playing with the same population every day, the re-matching procedure required them to wait in a virtual "waiting room" in between games, and the waiting room displayed a count of how many others were also waiting.

By noting when the waiting room emptied (i.e. to begin the next game) players could infer that roughly 50 people were playing at any given time. Moreover, they could also surmise from their own instructions that the population was very likely the same every day. Because players knew the length of the experiment, they could have reasonably concluded that they would encounter each other player an average of eight times in total, roughly once every three days. In turn, it is possible that the general expectation of repeated interactions could have elicited some form of generalized reciprocity, as the reviewer suggests. In the revised version, we have now clarified this aspect of the experiment and pointed out the possible limitation of our design.

That said, we do not think that this element of the game had much impact on player behavior, for three reasons. First, although players were never given misinformation or lied to, their attention was not drawn to this aspect of the game. Second, although our experiment ran for much longer than previous similar experiments, our subject pool was also 2 to 4 times larger; thus although the total number of repeat interactions was higher in our experiment, the frequency was lower (once every few days). And third, interactions were anonymous, and hence players could not condition their behavior on their knowledge of any specific partner. Accordingly, the self-reports show that very few players appear to have weighed this information in their thinking: of 38 players who were hand classified as CC only one mentioned the duration of the game (in addition one non--CC player mentioned repeated interactions with the same players).

3. While I am aware that there are some studies (e.g., Refs. 23 and 24 in the manuscript) that show that using Amazon Mechanical Turk is OK for experiments and gives similar results to the lab, others (sharing some authors) show large discrepancies [e.g, Rand et al., PNAS 111, 17093 (2014)]. Therefore, it is not clear to me whether this particular experiment would yield the same results in a lab. In particular, I see four possible problems here. First, given that the number of subjects is not exceedingly large, if some of them are playing under different names in the Turk this could induce a huge bias in the results. Can the authors check that this is not the case?

Although it is impossible to determine with complete certainty that no individual is running multiple Amazon Mechanical Turk accounts, it is extremely unlikely. First, owning multiple accounts is a clear violation of AMT terms of service, and detection by Amazon would likely result in termination of the user's account and forfeiting the earnings held in that account. This is an action that Amazon has taken in the past for a variety of infractions to their terms of service. Second, each AMT account is tied to an individual bank account and physical mailing address, and in many cases to a tax ID/SSN; thus fraudulently creating multiple accounts is nontrivial. Fourth, there is little benefit to a user in creating multiple accounts; thus it is highly unlikely that anyone would risk their entire income on Turk in order to game a single experiment. Finally, we kept track of individual IP addresses, finding only two players with the same address. Upon further investigation, the pair revealed themselves to be a married couple living together. Although this arrangement is not ideal and in future we would avoid it, we note that each member of the couple was assigned to a different subject pool, thus they never played one another. Moreover, they assured us via written communication that they worked independently and had not discussed their strategies. Although we cannot be certain of this assertion, we did not detect any unusual play from either player, nor did excluding their data affect our results.

Second, due to the fact that the experiment carries over during a month, it is perfectly possible that the subjects looked for info on the game they were playing, and therefore adapted their behavior accordingly. Can the authors be sure that the subjects remained "naive" during the experiment? Which consequences would the search for information have on the results?

We gave participants very clear instructions not to discuss the game with other Turkers or on forums. Requests not to discuss the details of experiments are common on Turk and our experience is that Turkers overwhelmingly honor such requests and even self-police this norm. Moreover, we actively monitored all relevant forums for the duration of the experiment and did not detect any discussion regarding strategy in the experiment. Thus (with the possible exception of the aforementioned married couple) we are confident that participants were not strategizing outside of the experiment itself.

Naturally we cannot rule out that participants recognized that the game was a form of PD and independently researched strategies on their own time. If they did this, however, they did not mention it in their exit surveys, nor did it affect their behavior in any obvious way. Moreover, it is not clear how casual online research would affect play: the Wikipedia entry on PD provides no clear guidance, nor does the academic literature; and searches generate links to other PD games. Thus although we expected our participants to think about their strategies and to adapt them over time in response to their experience—in other words *by design* they should cease to be naïve—we do not believe that online research into the PD itself had an important effect on the learning process. Of course, participants may have been influenced by other events happening in their lives, but that would be entirely within the scope of the effects that we intended to measure.

Third, I assume that of course the Turk identities involved in the experiment are the same all along, but can the authors be sure that the subjects are the same? Are there any tests or checks that aim to ensuring that the person is the same and is not leaving his/her account to somebody else? In a typical experiment one would not expect that this would be an issue, but in this long setup, it is perfectly possible that if somebody can't join one of the days, allows somebody else to play on his/her name.

Although it is impossible to know for certain that Turkers do not share their account credentials with other Turkers, we note that Turker accounts are also Amazon Payments accounts, which hold Turker earnings that can be used to purchase products and services on Amazon. Thus, giving another worker access to an account is similar to giving that worker access to one of your debit cards. Also as noted above, each Turk account is linked to a bank account and often to a Tax ID, as well as to a physical and email address. It is extremely unlikely that Turkers share such personally and financially sensitive information, nor is there any evidence from Turker forums or surveys to suggest that this kind of sharing takes place.

Finally, in the questionnaire players are asked whether they have participated before in similar games on the Turk. Was any action taken if the answer to this question was affirmative, i.e., the subject was excluded? Note that this is connected to my remark above about having information on the game or being "naive".

In the exit survey we asked participants to report the number of previous PD experiments in which they had participated. We show the results below. Although the modal response was zero

experience, more than half of respondents reported some experience, and a small fraction reported extensive experience. While we cannot compare these results with traditional lab studies, which as far as we are aware do not ask subjects to report prior experience, we suspect that the higher numbers are unreliable. We also suspect that the vast majority of “experiments” are single-shot PD games; thus even the most experience participants would have played many more rounds of PD (roughly 4000) in our experiment than prior to it. Nevertheless, it is clear that at least some Turkers had experience playing PD, as suggested by prior work [6]. To check that this prior exposure did not affect our results, we compared three groups of participants: those with zero self-reported experience (n=38), those who reported 1-4 prior experiments (n=28), and those who report 5 or more (n=28). Both in terms of overall cooperation rates and also the breakdown of CC vs. Threshold players, we found no significant differences between the populations; thus self-reported experience does not appear to affect our main results.

4. Still about the issue of the experimental setup, unless I overlooked something there is a lot of information missing about the setup. How are subjects paid? Do they get a exchange rate of the total number of points, are some games selected at random,...? All we know from the text (again, unless I'm wrong and if that's the case I'm sorry, but perhaps it needs to be more salient) is that they receive \$20 if they complete 18 sessions, as noted in the Methods section, but nothing else is said about the payment scheme. Another thing we do not know is the average payment the subjects get and, in connection with this, where are the subjects from. This is important because Turkers may be from many different countries and the payment should be appropriate to the country chosen and, if there are players from different countries, it is possible that payments are large for some and small for other. From the questionnaire (see p 10 of SI) it seems participants come from the US and Canada but it should be stated in the text and also how many come from each country, if this is indeed the case.

We have now provided all these details (see Results, pp. 6-9)

Finally, the authors should include the full instructions (as is customary in experimental economics) as shown to the participants so readers can check exactly what the subjects knew and how it was presented to them.

We have added the instructions to the SI.

5. Going now into the results, I find it remarkable the high level of cooperation one can see in Fig. 1,

particularly in panel F. It seems very large compared to the studies reported in the meta-analysis by Embrey et al. (Ref. 4 of the manuscript). If one looks at Table 1 in that paper, for values of g and l closer to the ones used here ($g=l=1$) one sees that the level of cooperation reported in the study AM1993 is considerably smaller, even in the last super game. What can be the reason for this? This is important because it appears (admittedly, maybe by chance) that there is a kind of trend in the results by some of the authors of this manuscript. For instance, looking at the paper by Wang, Suri and Watts (Ref. 17 of the manuscript) and comparing the results to similar papers by Gallo and Yan [PNAS 112, 3647 (2015)] and by Cuesta et al. [Sci. Rep. 5, 7843 (2015)] one sees that the cooperation levels are much higher in the former. Parenthetical remark: the authors might consider quoting these papers when mentioning reputation at the bottom of page 12, as they show that the cooperation observed on Ref. 17 arises from reputation.

We agree that the average level of cooperation that we observed is indeed high relatively to most previous work, although not without precedent. There are a number of reasons why our setup may have led to higher-than-typical cooperation.

First, although previous work [3, 7] has found that players recruited from MTurk cooperate at similar rates to those in lab studies, it is possible that the recent evolution of the MTurk community has resulted in a population that is more cooperative than the usual, also non-representative [8], population of subjects present in traditional lab experiments.

Second, prior work [9] has noted that cooperation rates in finitely repeated games are sensitive to choices in the game matrix parameters g and l , where lower values correspond to more cooperation. Given that our values $g=1$ and $l=1$ are at the low end of previous studies it is not surprising that we recover relatively high cooperation rates.

Third, prior work [9] has also shown that the duration of a finitely repeated game is highly predictive of initial cooperation levels. As our games were relatively long (10 rounds) compared with prior work, it is again not surprising that initial cooperation was relatively high. Moreover, analogous logic would suggest that the overall duration of the experiment could also be related to cooperation levels. Because our design required us to inform participants about the length of the experiment, this knowledge may have led to more cooperative behavior.

Finally, as noted in response to point 2 above, they could have inferred that they would meet the same players every few days, and that this general expectation of repeated interactions also facilitated cooperative behavior.

Although we do not believe that the high average level of cooperation changes our main results, we agree that it is a striking result and may raise concerns; thus in the revised paper we now address the matter explicitly.

6. While I very much like the classification of strategies (but see below under "Minor remarks") and the agent-based model designed on the basis of that classification, I believe that a more quantitative comparison of the results would be in order. For instance, on panel D in Fig. 4 (note the typo "resilent" in the horizontal axis label) we are shown a point corresponding to the experiment that is practically on top of the simulated line. However, if I understand correctly the line is an average, so

there should be a standard deviation interval around that line; error bars or shaded regions are needed in that plot. In addition, I believe that the comparison could involve the distribution of threshold strategies in the experiment and in the simulations: if these distributions were really similar that would be a very nice point for the simulation model.

The following plot shows the interquartile range (IQR) for the simulations. As should be clear, for the vast majority of value of alpha, the IQR is scarcely larger than the line thickness (the blocky nature of the window is due to the integral nature of the round of first defection). We do not find this extra information particularly helpful, thus for aesthetic reasons we would be inclined to leave the current figure as it is. If the reviewer thinks it is important to include the IQR, however, we would be happy to comply.

7. In the conclusions, on page 11, the authors state that their experiment supports the conclusion that roughly a 40% of the population behave altruistically. This is something that, provided all the points mentioned in this report are satisfactorily sorted out, I am ready to admit. In fact, I believe that the authors should comment on the "universality" of this result, namely that a fraction of people between 30% and 50% usually cooperate in the first round of prisoner's dilemmas or public goods games (see e.g. the review by Ledyard on the Handbook of Experimental Economics by Kagel and Roth (Princeton, 1995), and even on static networks, see the metaanalysis in Grujic et al. [Sci. Rep. 4, 4615 (2014)]. In connection with this, it is interesting to mention that recently Peysakhovich et al. [Nat. Commun. 5, 4939] and Capraro et al. [Sci. Rep. 4, 6790 (2014)] have found that comparable fractions of the population are consistently cooperative across games and across cost-benefit

ratios, and have even coined the term "cooperative phenotype" to describe these people. I believe that the authors should comment on this universal fraction of cooperators and on the possibility that these subjects are just born cooperators. Interestingly, a paper in the latest (at the time of writing this report) issue of PNAS by Yamagishi et al. [PNAS, Early Edition, May 2, 2016] seemingly connects altruistic giving in the DG with the thickness of the dorsolateral prefrontal cortex in the brain, which would point in the direction of some people being "natural cooperators" (cf. also my comment no. 1 above, this would be another alternative interpretation of the results).

We thank for the reviewer for these suggestions. We have now added a paragraph to the discussion that references this prior work and describes how our results add to it.

8. As a final suggestion, and given the demographics reported on Fig. S2, I believe that the authors should check how the cooperator percentage is distributed among male and female participants, and also among different age ranges (in this last case I believe that they could just try to split the age range in three intervals with approximately the same number of subjects, otherwise the statistics may be poor). Whether the results show differences in terms of gender or age or not, it is an interesting result in itself (the correlation between these variables and the threshold strategies would also be very interesting). Differences in cooperative behavior across genders are a controversial issue in the literature, with reports in favor and against, and age is also becoming of interest in recent years, so this could be an interesting contribution from this study without much effort. Of course, the caveat about the reliability of these data on the Turk remains, but if the authors trust their subject pool, I believe these analyses should be done.

Results for rates of resilient-CC play by gender and age are given below. No differences were significant. We have not added these results to the paper, but would be happy to add them to the SI if the reviewer feels they add value.

Minor remarks

1. On line 2 of the abstract, it is stated that "the evolution of cooperation in repeated games of prisoner's dilemma remains unresolved". This is the case only with finitely repeated games, there is no mystery in dyadic, infinitely repeated games, where infinite equilibria are possible and

experiments show abundantly that cooperation emerges through reciprocity. I am sure that the authors are well aware of this, they probably forgot to add "finitely", but they should add it in a revised version.

Thank you for raising this point. We have now modified the wording accordingly.

2. Regarding the location of the transition to a stationary behavior, the authors report that using a Kolmogorov Smirnov test to compare the distributions shows that they become effectively indistinguishable from day 7 on. Have the authors tried other tests applied to the distributions? And have they tried other criteria based on other magnitudes, such as, e.g., cooperation per round in the different days? Would any of these lead to different results?

Although the KS test is not perfectly suited to our case (it is designed for continuous distributions whereas ours is discrete), it is arguably better than the alternatives such as the Mann-Whitney U test, which is designed for ordinal numbers. Nevertheless, as a tentative robustness check we have conducted a MW test on the same data, with roughly similar results (see table below; p values greater than 0.05 highlighted).

In addition, the onset of a “stable” state at roughly day 7 can be inferred in at three other ways: first, by noting the change of slope in the cooperation rates for rounds 9 and 10 (Fig. 3A); second, by observing the cooperation rate over the course of the game (see Fig 4A), which at first changes from day to day but stabilizes after several days; and third, by observing the between-game “restart effect,” which rises for the first several days and then stabilizes (see Fig. 4B). Although these measures are less precise than the KS test applied to the distribution of round of first defection, they both yield similar results. Moreover, we emphasize that the precise day on which stabilization occurs is relatively unimportant as long as (a) it occurs well before the end of the experiment, and (b) it occurs well after day 1. Even if different tests yield slightly different estimates of the precise onset of the stable phase, they all agree on both these criteria.

After day	K-S statistic	K-S p-value	Mann-Whitney U	MW p-value
1	0.119886	0.000001	468179.0	0.000192
2	0.079265	0.004222	452340.5	0.069139
3	0.116710	0.000003	434550.5	0.000139
4	0.067663	0.021901	472798.0	0.416333
5	0.098592	0.000130	435855.0	0.000091
6	0.080558	0.003066	442348.5	0.000078
7	0.048479	0.192048	468674.5	0.075663
8	0.049679	0.179959	435583.0	0.004242
9	0.021636	0.978368	444000.5	0.330239
10	0.031956	0.709569	430857.0	0.025303
11	0.026316	0.893396	437976.0	0.115994
12	0.022215	0.973631	433163.0	0.330349

After day	K-S statistic	K-S p-value	Mann-Whitney U	MW p-value
13	0.017947	0.998458	410687.0	0.479973
14	0.012800	0.999999	413930.0	0.451055
15	0.017551	0.998843	411539.0	0.256666
16	0.039885	0.459107	402405.0	0.208344
17	0.046490	0.274213	400010.5	0.129383
18	0.026689	0.899757	406589.0	0.358326
19	0.042543	0.381455	390463.5	0.071810

3. On page 7, line 2 from bottom, the authors report that the ten threshold strategies plus the CC strategy account for "the vast majority of observed behavior". This should be properly quantified. They later indicate on p. 8 that roughly 60% play threshold strategies and roughly 40% play CC. Why are not the exact numbers given? And, more importantly, do the authors have any idea of what the remaining players do?

The plot below shows the fraction of strategies classified as "other" (i.e. not CC or Threshold) over the course of the experiment, by day. On day 1 almost 20% of strategies cannot be classified, as players switch back and forth between cooperate and defect during an initial learning period. However, we note that even this fraction falls to roughly 2% by day 5 and remains close to zero for the rest of the experiment.

4. The abstract is still formatted "Nature-style", as it appears that this manuscript has been transferred from Nature. This should be fixed and, given that the space restrictions are much more loose on Nature Communications, the authors should provide a good abstract, a proper introduction with a suitable revision of the literature, and all the necessary explanations and discussions of their setup and claims.

We have reformatted the paper to fit the Nature Communications style. Major changes include:

1. Shorter abstract
2. Separate sections for Introduction, Results, Discussion, and Methods
3. Longer introduction with more discussion of the literature
4. More details of the experimental designs, subject population, payments, etc.
5. Longer conclusion with more discussion of (a) relation with previous results, and (b) alternative explanations for observed behavior
6. More figures

Reviewer #3 (Remarks to the Author):

The authors report results from an online experiment on the finitely repeated prisoner's dilemma. The experiment was run for 20 days using Amazon Turk, holding the subject pool constant. On each day, subjects played multiple instances of the 10-round prisoner's dilemma (against changing opponents). As the length of a game is known, theory would predict that subjects should learn over time that they should not cooperate in the last round, after which they should learn not to cooperate in the second to last round, up to a point when there is no cooperation at all. However, according to the experiment, such an unraveling does not occur - the vast majority of subjects cooperates at least up to round 8. The authors argue that these high cooperation rates are due to the presence of "altruists", who would cooperate even in the very last round, provided their co-player cooperated in all previous rounds. Using individual-based simulations, the authors show that if roughly 40% of the players are altruists (which is the value suggested by the experiment), then the behavior of the remaining 60% can be explained as a rational response.

The setup of the experiment is impressive. To the best of my knowledge, the authors are first to provide data of play in the repeated prisoner's dilemma over multiple days. This kind of data is not only interesting from an experimental perspective - it should also be extremely useful to theorists who want to study human learning in social dilemmas.

The presented results are remarkable, and the research in general seems to be well executed. Somewhat unfortunately, however, the authors failed to provide some information that seems to be rather essential to fully judge the quality of the paper (concerning details of the experimental design and of the statistical methods). Provided these issues can be resolved, the manuscript certainly justifies publication in Nature Communications.

Major comments:

Quite essential information was missing. For example, I couldn't find information on the exact payoff values used in each one-shot game (i.e., the values of T, R, P, S); only the two derived quantities g and l have been reported on page 2. Without knowing these quantities, it is actually impossible to judge whether the reported results make sense, and whether they can be expected to be robust.

We apologize for the omissions. We have now specified the payoffs $T=7$, $R=5$, $P=3$, $S=1$ in the paper (see results, pp. 6-7).

Similarly, I would like to ask the authors to provide more information on (i) how much time the experiment took (i.e., how many hours did it take the subjects on average to participate in this experiment)

We now specify in the paper that sessions lasted an average of 35 minutes (most of them lasted between 30-40 minutes), corresponding roughly to between 10.5 and 11.5 hours of play, depending on how many sessions a player joined.

(ii) average earnings per participant over the whole experiment (differentiating between fixed compensations and variable compensations)

Participants made an average of \$4.47 per session (equivalent to an hourly wage of \$7.66) resulting in an average variable compensation of \$87.03 over the whole experiment. In addition, participants who completed at least 18 out of 20 sessions received a completion bonus of \$20.

(iii) average experience of the participants with previous social dilemma experiments

Please see response to Reviewer 2, point 3

Similarly, I could not find information on how exactly the statistical tests were performed. In particular, the authors need to explain which statistical models they have used, and how they have taken into account that the decisions of different individuals cannot be taken as fully independent, and that also different decisions of the same individual cannot be taken as independent.

Since partners are randomly assigned and anonymous we assume that different games in the same day are independent but we do not assume any independence between rounds of the same game which we now state on page 12.

I was somewhat surprised about the relatively high cooperation rates in this experiment. Already in the very first round of the first game on day 1, subjects seem to cooperate with almost 90% probability. In the laboratory experiments on the prisoner's dilemma I know of, initial cooperation rates are typically much lower (see e.g. Ref. 3, Ref. 14, Hilbe, Röhl, Milinski, Nature Communications 2013; Xu, Zhou, Lien, Zheng & Wang, Nature Communications 2016). I would like the authors to comment on this issue - are the high cooperation rates a consequence of the experimental design, of the chosen payoff values, or of the fact that Amazon Turk was used?

Please see response to Reviewer 2, point 5.

When classifying the players' behaviors, the authors only allow for strategies that have the property that if the strategy prescribes to defect in round r , the strategy also prescribes to defect in all subsequent rounds. How often did the authors observe behavior that was inconsistent with this property (i.e., games in which a player defected in one round but cooperated in the next).

Please response to Reviewer 2, minor remark 3.

Papers in Nature Communications should have a formal Introduction and a Discussion section. Given the rather multidisciplinary readership of Nature Communications, the authors should make use of the Introduction section to explain in more detail what the "altruism hypothesis" and what the "rationality hypothesis" is (I am afraid many readers will not know the key references 1-4). Also, given the multidisciplinary scope of NatComms, it may be a good idea to have a somewhat broader bibliography that also covers results from biology, mathematics and psychology (at the moment, most of the articles cited have an economics background).

As noted in our responses to both reviewers 1 and 2 we have reformatted the paper for Nature Communications, including a formal introduction and discussion. As requested, the introduction now includes a more detailed description of the rational cooperation hypothesis. Here we note, that in light of the comments of reviewer 2, we no longer refer to CC players as altruists, preferring instead to list altruism as one of four possible explanations for their behavior. Correspondingly we no longer make a distinction between the "rationality" and "altruism" hypotheses, instead referring just to the "rational cooperation" hypothesis and noting that its prediction for long-run cooperation depends on the frequency and resilience of conditional cooperators. Finally, we have added many more citations to related work, almost doubling the length of our bibliography.

Minor comments:

(-) The variable g is used for two different purposes, to denote a particular payoff quantity (on page 2), and to refer to an instance of a game (e.g. on page 3).

Thank you for pointing this out. We have now changed the game index to j .

(-) Coming from the evolutionary game theory literature, I find the term "altruists" somewhat unfortunate - most researchers in my field will associate altruists as people who cooperate in every single round, irrespective of the previous history of play. Thus I would recommend to use "conditional cooperator" or "grim trigger" instead, or at least to clarify the intended meaning of the word "altruist".

Our use of the term "altruism" comes from the economics literature, where it generally refers to other-regarding preferences (see e.g. Andreoni and Miller 1993), a considerably more encompassing definition than ALL C. (as an aside, we do not observe any ALL C players, nor are we aware of ALL C strategies being observed in other comparable experiments).

As we note in our response to Reviewer 2, however, although the self-reports do indicate that at least some CC players were motivated by other-regarding preferences, hence could legitimately be referred to as altruists, the evidence is less clear in other cases. For example, a number of players invoked fairness as a reason for their behavior, but it is not clear whether their desire to be fair was altruistic in nature or simply reflected their internalization of a norm (see also Reviewer 4). To avoid potential confusion, therefore, we now refer to CC players as "resilient cooperators," a classification that we believe can be sustained on behavioral grounds alone, and instead raise altruism as one of four possible explanations for the observed behavior.

Finally, why invoke a new term (“resilient cooperator”) when an existing term, such as conditional cooperator or grim trigger player, could also describe the behavior? The reason is that a critical distinguishing feature of the players that we classify as CC is precisely that they remain CC throughout the experiment, in spite of costly exploitation by Threshold players. Conversely, many players whom we ultimately classify as Threshold players would be indistinguishable from CC players based on one or even a few days’ play. Resilience being every bit as much a defining feature of our CC players as the conditional nature of their cooperation, simply calling them “conditional cooperators” would not, we believe, accurately communicate their behavior; thus we prefer the novel term “resilient cooperator.”

(-) Page 7: It is not immediately clear why the fact that between 15 and 20% of the games ended with full cooperation implies that 40% of the subjects would always cooperate until the co-player defected. Please explain in more detail.

Our reasoning is as follows: 16% of games display cooperation all the way through round 10; the only way that cooperation can continue for all 10 rounds is if both players are playing CC (equivalently, Grim Trigger); pairs of players are randomly matched for each game, hence if fraction x of players are CC then $x^2 = 0.16$, or $x = 0.4$. We added this explanation to page 15.

(-) In the abstract and on page 11, the authors say that “the presence of altruists is both necessary and sufficient for cooperation to sustain itself” - I find this formulation somewhat inappropriate, as it pretends mathematical accuracy. Also, it seems to me that while altruists certainly help to sustain cooperation, the CC strategy described in the main text is not the only way how one could uphold cooperation.

This is a fair point. We agree and have softened the language appropriately.

(-) Figure S6: what would happen for larger beta values, e.g. beta=0.5 or beta =1? At the moment, the authors are entirely focusing on the case of “strong selection” (where players would most often adopt the best strategy), whereas researchers in evolutionary game theory are sometimes also interested in the case of “weak selection”.

The reviewer is correct that we cannot be sure of the exact value of β . To check for robustness, therefore, we ran the simulations for β varying between 0.001 and 1. As we show in the figure below, the results are extremely robust over two orders of magnitude $0.001 \leq \beta \leq 0.1$. For larger values of β still, including 0.5 and 1, we do see qualitatively different results; however, we note that these values of β are extremely high, corresponding to players updating their strategies almost every round which is much faster than our human subjects. To clarify, we have also included the strategy heat maps for $\beta = 0.5$ and 1. Comparing these heat maps with the empirical heat map, it is clear that these values are far from realistic. Thus to the extent that the true value of β differs from our assumed value (0.005), we do not believe it will affect our qualitative conclusions.

Reviewer #4 (Remarks to the Author):

Report for "Altruists stabilize long-run cooperation in the finitely repeated Prisoner's Dilemma" by Andrew Mao, Lili Dworkin, Siddharth Suri and Duncan J. Watts

Summary:

This paper presents the results of an experiment that studies cooperative behavior in a finitely repeated prisoner's dilemma (PD). Subjects, recruited via Amazon's Mechanical Turk, play a series of 10 round finitely repeated PDs over 20 sessions (each held at a different day). In each session, subjects play 20 separate finitely repeated PDs. Subjects remain anonymous throughout the experiment and are randomly assigned to new partners between games.

The experimental design allows the authors to study long-term behavior in this context. Contrary to standard theoretical predictions, main results show partial cooperation to stabilize after a limited period of unravelling (first 7 sessions). The analysis suggests a significant portion of the population to follow a conditionally cooperative strategy that never preempts defection. Simulation results resulting from a learning model (where people update their beliefs about the distribution of strategies used by the population) show that the presence of such subjects can rationalize the data and explain the partial unravelling in cooperation.

Comments:

The paper's main contribution to the literature is to directly study long term dynamics. Since subjects participate in 20 sessions of 20 finitely repeated PDs, there is direct experimental evidence of how subjects behave in such an environment after $20 \times 20 = 400$ distinct individual experiences with the game. The results are clear as further unravelling of cooperating looks convincingly unlikely in this context. However, I have several concerns with the interpretation of the results.

Most importantly, controlling for emergence of social norms and community enforcement (as in Kandori 1992) is an issue here given that a fixed number of subjects are repeatedly matched with each other over the course of a long experiment. Note that there are 94 subjects who are randomly matched with each other for 400 finitely repeated games. This implies that on average any two subjects interact roughly about 4 times. This can create dynamic collective reinforcement. I think there is some evidence that at least some subjects recognize this effect and choose their strategy accordingly. First, Figure 1 suggests Round 8 cooperation to decline slightly in the last few sessions of the experiment. Second, subject questionnaire responses at the end of the experiment indicate this type of thinking. I show some below:

#5: "I figured I might as well try to get others to adopt a better strategy. . . and more people started going with 5 and 5 all the way through."

#42: "I felt going beyond this was idiotic because in the end to continue in this fashion you are jeopardizing the whole groups pay."

#43: "Knowing that we were playing the same participants every day, I tried to learn what patterns others were playing so I could adjust my play to benefit me but still be fair."

#45: "It stayed mostly the same, I cooperated more than I thought I would. I guess I kind of hoped it would encourage others to continue to cooperate more as well."

#54: "I started out by trying to be cooperative. . . I was concerned that the more I defected, the more mistrust would seep into the game and the worse everyone would do (though my concern was with my own results, not others'). . . I was willing to lose a few pennies each game if it meant people cooperated for 8 or 9 rounds at least..."

I think it's important for the paper to address this concern. In several sections of the paper, it is stated that long term behavior strikingly appears to be stable in the absence of reinforcement mechanisms such as reputation or punishment. However, such reinforcement mechanism might be at work in this dynamic context. The key question is how much more unravelling would we expect to observe if subjects knew they would never interact with the same person, or further they would never interact with someone who interacts with this person in the future?

First, we agree with the reviewer that the presence of social norms could account for some of the cooperation that we observe. As we note in our responses to Reviewers 2 and 3, for example, a number of participants invoked fairness as an explanation for their behavior in their self-reports, while others mentioned feeling guilty for having defected first. While previously we had interpreted these responses as evidence for altruism, on reflection they are equally consistent with the internalization of social norms. Therefore, as noted above, we now refrain from positing any particular interpretation of the CC players' behavior, labeling them instead "resilient cooperators"

and raising four possible explanations for their behavior in the discussion section, including both altruism (i.e. other-regarding preferences) and the internalization of social norms. We also refer more extensively to the literature on social norms in games of cooperation. We thank the reviewer for raising this important alternative explanation.

Second, we wish to draw a distinction between social norms that players bring with them into the experiment and norms that develop within the experiment itself. As just explained, at least some observed behavior does appear to be explained by the first type of norm, which is also the focus of the aforementioned literature. We also considered the second type of norm—e.g. a “rule” that emerges during the experiment of the form “it is OK to defect in the last round or two but not before.” Such a rule could also account for the stabilization of unraveling that we observed, and at first we suspected that that was what had happened. It was to test for this hypothesis, in fact, that we constructed the agent-based model. Recall that the model invoked only two types of players: resilient CC players, and “rational” players who selfishly best-respond to the inferred distribution of strategies in the population. In other words, while the model implicitly allowed for norms to be imported (via the CC players) it did not allow for the emergence of any novel norm-like rules. If the observed cessation in unraveling was due to an emergent norm, then the model would have failed to account for it. That the model did in fact account both the observed unraveling and its cessation therefore is evidence in favor of the null (i.e. no emergent norms) hypothesis.

Finally, a second distinction is between norm-based behavior and what might be termed “long-run” self-interest, which could arise when players decide to cooperate on the grounds that they will be better off selfishly if everyone cooperates. As we now discuss in the paper, long-run self-interest is yet another possible interpretation of the observed behavior. Moreover, as we note in our response to Reviewer 2, the prospect of repeatedly interacting with the same players could have increased the salience of this particular motivation vis-à-vis a variant of the experiment in which players believed they would never encounter the same partner twice. As we also noted in our response to Reviewer 2, it is difficult to estimate how much the prospect of repeated interactions affected overall cooperation; however, there are some reasons to think that the effect is not that large.

First, players were never explicitly told that they would be playing with the same population every day, nor were they told the size of the total population. Thus although they could have inferred that they belonged to a stable population of roughly 50 they could not have been certain of it, nor was their attention drawn to this aspect of the game. Reflecting this design choice, the self-reports show that the majority of cooperative players do not appear to have weighed this information heavily in their self-reported thinking. On this point, we note that the reviewer’s next point (regarding the session restart effect) also points to the likely lack of salience of repeated interactions.

Second, even if players had a general expectation of repeated encounters, the strict anonymity of the game would have prevented them from knowing which player they were encountering in any particular game. Thus it was impossible for players to establish any form of reputation, or to condition their behavior on their partner’s previous behavior.

Finally, we note that to the extent that cooperative behavior can be explained by long run self interest (i.e. players felt that they would do better personally if everyone cooperated),

the reasoning depends only on the general expectation that other players will reason the same way; it does not require that the same players are encountered repeatedly.

Thus although, as we concede above, the repeated nature of the interactions likely increased cooperation to some degree, we do not believe that this particular feature of our design was critical to our results. Naturally, however, this is a speculative hypothesis and we hope that a future experiment will test it. We have added a sentence to our discussion section to reflect this possible limitation of our design.

Session restart effects are very dramatic. Can the authors provide more insight on this? Could subjects be treating every session to be independent? The subjects might mistakenly believe that they are playing against a new group of people. Or it would be sufficient for them to believe that a significant portion of the subject pool makes such a mistake.

As noted in the previous response, players were not explicitly told that their pool of players would remain constant from day to day; however, they could have inferred as much from their own instructions, which assigned them to a particular time of day. Other than this, we do not have a good explanation for the session restart effect. We note, however, that the game restart effect, which is a consistent feature of finitely repeated games that are themselves repeated and is even larger in magnitude than the session restart effect, has also not been adequately explained. Possibly the best explanation—not entirely satisfactory—is that players simply “reset” when starting a new game/session, whereas they do not when playing the same number of rounds/games in a single game/session. We have now quantified this effect in the revised version (see Fig. 4C), but a satisfying theoretical explanation awaits future work.

Then the learning effects cannot carry through across different sessions. (There is also a question about how the learning model accounts for the restart effects that was not clear in the paper.) How do we interpret behavior in the last session in light of the restart effects? Is it long run as in after 400 repetitions of the game, or long run as in just 20 repetitions of the game. An interesting exercise would be to repeat a portion of this experiment where subjects play, for example, 80 finitely repeated PDs in one session. Play in the last repeated game here can compared to play at the end of the 4th session in the original experiment. This should be indicative of to what extent results of this paper can be interpreted as "long-term" behavior.

We agree that this would be an interesting variation, however it would require an entire new experiment; thus we believe it would be more appropriate for future work.

1. Watts, D.J., *Small Worlds : The Dynamics of Networks Between Order and Randomness*. 1999, Princeton: Princeton University Press.
2. Hanaki, N., et al., *Cooperation in evolving social networks*. *Management Science*, 2007. **53**(7): p. 1036-1050.
3. Suri, S. and D.J. Watts, *Cooperation and contagion in web-based, networked public goods experiments*. *PLoS One*, 2011. **6**(3): p. e16836.
4. Wang, J., S. Suri, and D.J. Watts, *Cooperation and assortativity with dynamic partner updating Supporting Information*. 2012.

5. Mason, W., S. Suri, and D.J. Watts. *Long-run learning in games of cooperation*. in *Proceedings of the fifteenth ACM conference on Economics and computation*. 2014. ACM.
6. Chandler, J., P. Mueller, and G. Paolacci, *Nonnaïveté among Amazon Mechanical Turk workers: Consequences and solutions for behavioral researchers*. *Behavior research methods*, 2014. **46**(1): p. 112-130.
7. Horton, J.J., D.G. Rand, and R.J. Zeckhauser, *The online laboratory: Conducting experiments in a real labor market*. NBER Working Paper, 2010.
8. Henrich, J., S.J. Heine, and A. Norenzayan, *The weirdest people in the world?* *Behavioral and brain sciences*, 2010. **33**(2-3): p. 61-83.
9. Embrey, M., G.R. Fréchette, and S. Yuksel, *Cooperation in the finitely repeated prisoner's dilemma*. Available at SSRN 2743269, 2015.

Reviewers' comments:

Reviewer #1 (Remarks to the Author):

While I appreciate the attempts of the authors to revise their manuscript in response to the referee comments, I have further comments in response to the unsatisfactory, and in part seriously ambiguous, replies of the authors.

Authors seem to believe that learning dynamics in evolutionary games is something entirely different from copying a more successful opponent. That is not the case, especially not in humans, where learning whom to imitate and why is just as well a learning process as it is to internally learn to adopt another strategy. In theoretical research, this is routinely dealt with by considering myopic strategy updating instead of imitation. While the results are sometimes different, this does not excuse the neglect of research concerning imitation. If anything, the authors should do much better to discuss the similarities between imitation and learning, and in fact point out the very real complementary aspects of just how humans adopt new strategies. A useful reference is Exploration dynamics in evolutionary games, PNAS 106, 709-712. Besides, research concerning specifically learning effects is also not hard to come by. Admittedly, it is less than for imitation, but if one is to suggest a new learning model as the authors do, previous same attempts, especially the most recent ones, should be acknowledged. One of the key results of the theoretical part of this research, as highlighted in the abstract, namely that "using a standard learning model we predict that the presence of more than a critical fraction of resilient cooperators can permanently stabilize unraveling among a majority of rational players" was demonstrated in Directional learning and the provisioning of public goods, Scientific Reports 5, 8010. The key effects of learning for human cooperation have also been clearly pointed out before in Learning dynamics explains human behaviour in Prisoner's Dilemma on networks, Journal of The Royal Society Interface 11, 20131186.

Some of the arguments as to why the introduction was originally deficient are incredibly thin. Your paper was submitted to Nature Communications and evaluated as such. You had an opportunity to revise the manuscript accordingly prior to agreeing to the transfer from Nature. Also, just because some of your earlier references were cited in some of the newly suggested papers does not make a poor introduction acceptable. Research is moving on, and of course newer papers cite older papers, but that does not mean that it is fine to just cite older papers and neglect new ones. The argument is among the most broken ones I have had the pleasure of reading as an excuse for a poor introduction. The introduction is a bit better now, but several newly added references are missing publication details, and it is difficult to give credit to research if even such elementary aspects of the work contain that many errors.

So many similar experiments have been conducted and published recently, much of which still ignored by the authors, that it is impossible for me to understand how this work makes a seminal contribution worthy of Nature Communications. Experiments and theoretical research have shown before that only a few cooperators (now called resilient cooperators by the authors), and even if emerging just by chance, are enough to revert a decline to full defection under learning, and in some circumstances may be enough to lead the population to a state where everybody cooperates at the end. It is nice that the experiments of the authors confirm this, but in the light of existing research on the same subject, the presented result are underwhelming. The fact that the authors insist on their novelty and refuse to integrate it properly with existing recent research, in part with quite horrific excuses, is unacceptable.

Reviewer #2 (Remarks to the Author):

The authors have satisfactorily answered most of my questions and have modified their

manuscript to make it more readable and also to give more details about their experiment and its interpretation. I believe that the manuscript has improved very much and it should be ready for publication once the authors take care of a few remaining points:

1. In their response to my point 5, the authors state:

"Although we do not believe that the high average level of cooperation changes our main results, we agree that it is a striking result and may raise concerns; thus in the revised paper we now address the matter explicitly."

I may have missed it (and if that is the case, I apologize in advance) but I haven't found this discussion in the revised version, and I do believe it should be included because it's a relevant point.

2. Also about point 5, in my previous report I indicated two references that the authors should quote in connection with high levels of cooperation. The authors have included one, the paper by Gallo and Yan, and excluded the other without any explanation, the one by Cuesta et al. This must be corrected and the paper by Cuesta et al must be properly cited, in so far as it is equally relevant as the paper of Gallo and Yan (in fact, these authors acknowledged exchanges from Cuesta et al in their paper) and, along with the (also cited in the paper, Ref. 15) one by Corten et al, form the core of knowledge on this matter. As Nature Communication does not have any policy about a maximum number of references, it is not correct to exclude relevant works from them.

3. Going on on the subject of references, on my point 7 I mentioned a number of them that the authors could benefit from in the introductions. The authors answer: "We have now added a paragraph to the discussion that references this prior work and describes how our results add to it." However, only one of the references I indicated is cited (Peysakovich et al). While this is not as serious an omission as the previous one, because there are very many works that can be cited in an introduction and one must of course choose, I find it peculiar that the rest are not cited. In any event, in this case I am willing to accept the authors' criterion, but I want to draw this point to their attention for consideration.

4. About the same topic as Peysakovich et al, namely the existence of "preprogrammed phenotypes", during the revision process a very relevant reference came out, namely " Humans display a reduced set of consistent behavioral phenotypes in dyadic games", Poncela-Casasnovas et al, Science Advances 05 Aug 2016: Vol. 2, no. 8, e1600451, DOI: 10.1126/sciadv.1600451, that the authors might want to look at in so far as it also bears on their discussion of the phenotype point.

5. Finally, I have very much liked the various discussions and additional data and plots given by the authors in their answer to my comments. I believe that they are very useful and contribute to improve the quality of the manuscript and, as the authors themselves offer to do, I would ask them to include all those results in the supporting information, which should not be a lot of work as plots and comments are already prepared.

With these revisions, I am confident that the final version of the paper will be acceptable for Nature Communications.

Reviewer #3 (Remarks to the Author):

In the revised manuscript, the authors have taken all my previous comments into account. While I think that this is now a fine manuscript, I would appreciate if the authors could take the following suggestions into account when preparing the final version of the paper.

Optimally, the abstract should start with 1-3 introductory sentences that explain the background of the experiment, including the research question tackled with this experiment.

I feel that Figs. 1 and 2 should be transferred to the SI — both figures are not essential to the understanding of the experiment's results, and it's actually difficult for the reader to learn anything from these figures. If the authors want to present the basic setup of the experiment, I would rather suggest that they prepare a simple schematic illustration, for example like Fig. 1 in "Cooperating with the Future" of Hauser et al. (Nature 2014).

I am still somewhat irritated by the high cooperation rates in this experiment (note that Ref. 28 should not be used to justify these high cooperation rates, as done on page 9, because also in Ref. 28 initial cooperation rates are much lower, starting at ~30-50%).

The authors argue that their high cooperation rates may be due to

- (1) the subject pool in MTurk,
- (2) the payoff parameters ($g=1$, $l=1$) which are supposed to be cooperation-friendly,
- (3) The long duration of each game (10 rounds), and that many games are played within the same population.

At least for the experiments pertaining to indefinitely repeated games, it seems to me that 10 rounds is rather at the low end of considered round numbers in the literature. Moreover, for any experiment that is based on the Axelrod tournament (with $T=5$, $R=3$, $P=1$, $S=0$) it follows that $g=1$ and $l=1/2$, and hence such an experiment would be even more cooperation-friendly — yet I have never seen such high cooperation rates in the lab before (especially not in the very first rounds).

I don't think there is very much the authors could do about this; but maybe they could add another disclaimer to the discussion that reminds the reader that the cooperation rates presented here are somewhat high, and that future experiments could help to clarify this point.

Also, I would appreciate if the authors could add a statistical analysis to their SI, in which they discuss how initial cooperation rates of the players depend on the MTurkers' experience with social dilemma experiments (i.e. whether more experienced players are more or less cooperative in the first round of the first game than naive players).

Page 3: "prior work has reached mixed conclusions regarding the long run fate of cooperation in repeated games" —> "... cooperation in FINITELY repeated games".

Reviewers' comments:

Reviewer #1 (Remarks to the Author):

While I appreciate the attempts of the authors to revise their manuscript in response to the referee comments, I have further comments in response to the unsatisfactory, and in part seriously ambiguous, replies of the authors.

Authors seem to believe that learning dynamics in evolutionary games is something entirely different from copying a more successful opponent. That is not the case, especially not in humans, where learning whom to imitate and why is just as well a learning process as it is to internally learn to adopt another strategy. In theoretical research, this is routinely dealt with by considering myopic strategy updating instead of imitation. While the results are sometimes different, this does not excuse the neglect of research concerning imitation. If anything, the authors should do much better to discuss the similarities between imitation and learning, and in fact point out the very real complementary aspects of just how humans adopt new strategies. A useful reference is Exploration dynamics in evolutionary games, PNAS 106, 709-712. Besides, research concerning specifically learning effects is also not hard to come by. Admittedly, it is less than for imitation, but if one is to suggest a new learning model as the authors do, previous same attempts, especially the most recent ones, should be acknowledged. One of the key results of the theoretical part of this research, as highlighted in the abstract, namely that "using a standard learning model we predict that the presence of more than a critical fraction of resilient cooperators can permanently stabilize unraveling among a majority of rational players" was demonstrated in Directional learning and the provisioning of public goods, Scientific Reports 5, 8010. The key effects of learning for human cooperation have also been clearly pointed out before in Learning dynamics explains human behaviour in Prisoner's Dilemma on networks, Journal of The Royal Society Interface 11, 20131186.

To clarify, the model that we use to recover the behavior observed in our experiment does not rely on imitation. Rather, it is a "smoothed fictitious play" model: agents infer the distribution of strategies in the population from their past interactions, and then select their strategy by computing expected utilities of all available strategies against the currently inferred distribution and stochastically best-responding. Thus our model is intrinsically forward looking and optimizing rather than backward looking and imitative. As a result it differs in important ways from the models referenced above. For example, the threshold rules of the sort that dominate play in finitely repeated games require forward-looking agents who are anticipating the end of the game and who are seeking to exploit it; thus the strategy space for forward looking agents playing finitely repeated games is inherently different from those of backward looking agents playing one shot or indefinitely repeated games. More generally, the whole question that we investigate—what happens to these thresholds as players gain experience—depends on forward looking optimization. In that sense, it is indeed different from "learning" in evolutionary games. We have edited the second paragraph of our introduction to further clarify this distinction.

Some of the arguments as to why the introduction was originally deficient are incredibly thin. Your paper was submitted to Nature Communications and evaluated as such. You had an opportunity to revise the manuscript accordingly prior to agreeing to the transfer from Nature. Also, just because

some of your earlier references were cited in some of the newly suggested papers does not make a poor introduction acceptable. Research is moving on, and of course newer papers cite older papers, but that does not mean that it is fine to just cite older papers and neglect new ones. The argument is among the most broken ones I have had the pleasure of reading as an excuse for a poor introduction. The introduction is a bit better now, but several newly added references are missing publication details, and it is difficult to give credit to research if even such elementary aspects of the work contain that many errors.

No excuse was intended. Nature offers an automated service for transferring manuscripts between journals, and we took advantage of this service. We regret that this was not clear initially, but we have now clarified it. If the reviewer could point us to specific errors in the bibliography we would be happy to correct them.

So many similar experiments have been conducted and published recently, much of which still ignored by the authors, that it is impossible for me to understand how this work makes a seminal contribution worthy of Nature Communications. Experiments and theoretical research have shown before that only a few cooperators (now called resilient cooperators by the authors), and even if emerging just by chance, are enough to revert a decline to full defection under learning, and in some circumstances may be enough to lead the population to a state where everybody cooperates at the end. It is nice that the experiments of the authors confirm this, but in the light of existing research on the same subject, the presented results are underwhelming. The fact that the authors insist on their novelty and refuse to integrate it properly with existing recent research, in part with quite horrific excuses, is unacceptable.

We are not aware of any experiments in which human subjects have played finitely repeated games of PD over the course of many days. As the behavior of interest only manifested itself over the course of several days we are confident that our results are also novel.

Reviewer #2 (Remarks to the Author):

The authors have satisfactorily answered most of my questions and have modified their manuscript to make it more readable and also to give more details about their experiment and its interpretation. I believe that the manuscript has improved very much and it should be ready for publication once the authors take care of a few remaining points:

We are grateful for the reviewer's earlier questions and happy that we have largely satisfied them. Responses to remaining issues are detailed below.

1. In their response to my point 5, the authors state:

"Although we do not believe that the high average level of cooperation changes our main results, we agree that it is a striking result and may raise concerns; thus in the revised paper we now address the matter explicitly."

I may have missed it (and if that is the case, I apologize in advance) but I haven't found this discussion in the revised version, and I do believe it should be included because it's a relevant point.

On pages 8-9 of the revised manuscript we added a slightly condensed version of our response to your point 5, namely:

"There are a number of reasons why our setup may have led to overall higher-than-typical cooperation. First, although previous work [Suri and Mason 2011, Horton et al. 2011] has found that players recruited from MTurk cooperate at similar rates to those in lab studies, it is possible that the recent evolution of the MTurk community has resulted in a population that is more cooperative than the usual, also non-representative [Henrich et al. 2010], population of subjects present in traditional lab experiments. Second, prior work [Embrey et al. 2015] has noted that cooperation rates in finitely repeated games are sensitive to choices in the game matrix parameters g and l , where lower values correspond to more cooperation. As noted above, our values $g=1$ and $l=1$ were at the low end of previous studies, thus it is not surprising that we recover relatively high cooperation rates. Third, prior work [Embrey et al. 2015] has also shown that the duration of a finitely repeated game is highly predictive of initial cooperation levels. Our games, which were 10 rounds long, were relatively long compared to previous experiments; thus once again it is not surprising that cooperation levels were relatively high. Moreover, analogous logic would suggest that the overall duration of the experiment could also be related to cooperation levels. Because our design required us to inform participants about the length of the experiment, this knowledge may also have led to more cooperative behavior. Finally, although players were not explicitly told the size of the population with whom they were being matched, they could have inferred this information from the counter in the virtual waiting room. Likewise, they were not directly informed that they were playing with the same population every day but could have inferred as much from their instructions, and hence could have reasonably concluded that they would anonymously encounter the same players several times over the course of the experiment. It is plausible, therefore, that the general expectation of repeated interactions also facilitated cooperative behavior."

We hope this discussion is satisfactory.

2. Also about point 5, in my previous report I indicated two references that the authors should quote in connection with high levels of cooperation. The authors have included one, the paper by Gallo and Yan, and excluded the other without any explanation, the one by Cuesta et al. This must be corrected and the paper by Cuesta et al must be properly cited, in so far as it is equally relevant as the paper of Gallo and Yan (in fact, these authors acknowledged exchanges from Cuesta et al in their paper) and, along with the (also cited in the paper, Ref. 15) one by Corten et al, form the core of knowledge on

this matter. As Nature Communication does not have any policy about a maximum number of references, it is not correct to exclude relevant works from them.

We apologize for the omission, which was unintentional, and have now cited Cuesta et al.

3. Going on on the subject of references, on my point 7 I mentioned a number of them that the authors could benefit from in the introductions. The authors answer: "We have now added a paragraph to the discussion that references this prior work and describes how our results add to it." However, only one of the references I indicated is cited (Peysakovich et al). While this is not as serious an omission as the previous one, because there are very many works that can be cited in an introduction and one must of course choose, I find it peculiar that the rest are not cited. In any event, in this case I am willing to accept the authors' criterion, but I want to draw this point to their attention for consideration.

We had in fact cited Ledyard (1995) in addition to Peysakovich et al., but in a different location. We have now added additional citations to Ledyard, and added Capraro et al (2014) and Grujic et al (2014). We decided not to add Yamagishi (2016) as this seemed less relevant.

4. About the same topic as Peysakovich et al, namely the existence of "preprogrammed phenotypes", during the revision process a very relevant reference came out, namely " Humans display a reduced set of consistent behavioral phenotypes in dyadic games", Poncela-Casasnovas et al, Science Advances 05 Aug 2016: Vol. 2, no. 8, e1600451, DOI: 10.1126/sciadv.1600451, that the authors might want to look at in so far as it also bears on their discussion of the phenotype point.

We have added a reference to this paper as part of the same discussion.

5. Finally, I have very much liked the various discussions and additional data and plots given by the authors in their answer to my comments. I believe that they are very useful and contribute to improve the quality of the manuscript and, as the authors themselves offer to do, I would ask them to include all those results in the supporting information, which should not be a lot of work as plots and comments are already prepared.

We have added these plots and related text to the SI as requested.

With these revisions, I am confident that the final version of the paper will be acceptable for Nature Communications.

Reviewer #3 (Remarks to the Author):

In the revised manuscript, the authors have taken all my previous comments into account. While I think that this is now a fine manuscript, I would appreciate if the authors could take the following suggestions into account when preparing the final version of the paper.

We are grateful for the reviewer's previous comments and are happy that our revised manuscript has taken them into account. We have responded to the reviewer's remaining concerns below.

Optimally, the abstract should start with 1-3 introductory sentences that explain the background of the experiment, including the research question tackled with this experiment.

Noting that we are already straining the abstract word limit for Nature Communications, we have added an initial intro/framing sentence.

I feel that Figs. 1 and 2 should be transferred to the SI — both figures are not essential to the understanding of the experiment's results, and it's actually difficult for the reader to learn anything from these figures. If the authors want to present the basic setup of the experiment, I would rather suggest that they prepare a simple schematic illustration, for example like Fig. 1 in "Cooperating with the Future" of Hauser et al. (Nature 2014).

We have transferred Figs. 1 and 2 to the SI, as requested.

I am still somewhat irritated by the high cooperation rates in this experiment (note that Ref. 28 should not be used to justify these high cooperation rates, as done on page 9, because also in Ref. 28 initial cooperation rates are much lower, starting at ~30-50%).

The authors argue that their high cooperation rates may be due to

- (1) the subject pool in MTurk,
- (2) the payoff parameters ($g=1$, $l=1$) which are supposed to be cooperation-friendly,
- (3) The long duration of each game (10 rounds), and that many games are played within the same population.

At least for the experiments pertaining to indefinitely repeated games, it seems to me that 10 rounds is rather at the low end of considered round numbers in the literature. Moreover, for any experiment that is based on the Axelrod tournament (with $T=5$, $R=3$, $P=1$, $S=0$) it follows that $g=1$ and $l=1/2$, and hence such an experiment would be even more cooperation-friendly — yet I have never seen such high cooperation rates in the lab before (especially not in the very first rounds).

I don't think there is very much the authors could do about this; but maybe they could add another

disclaimer to the discussion that reminds the reader that the cooperation rates presented here are somewhat high, and that future experiments could help to clarify this point.

We had added a sentence to the final paragraph of the discussion noting the discrepancy and calling for further clarifying work. We have also removed ref 28.

Also, I would appreciate if the authors could add a statistical analysis to their SI, in which they discuss how initial cooperation rates of the players depend on the MTurkers' experience with social dilemma experiments (i.e. whether more experienced players are more or less cooperative in the first round of the first game than naive players).

We have added a figure in the SI showing cooperation rates on the first round of the first game, as well as averaged over games on the first day and over the whole experiment, grouped by self-reported player experience. We detected no statistically significant differences between the behavior of these groups.

Page 3: "prior work has reached mixed conclusions regarding the long run fate of cooperation in repeated games" —> "... cooperation in FINITELY repeated games".

Thanks – corrected.

REVIEWERS' COMMENTS:

Reviewer #1 decided not to leave any comment, largely retaining their previous opinion on the manuscript.

Reviewer #2 (Remarks to the Author):

I appreciate the effort made by the authors in addressing the points I raised in my last report, and I am now fully satisfied. I am sure that, in this version, it is a very nice and interesting paper, that of course deserves publication in Nature Communications, and I believe it will give rise to many productive discussions.

Reviewer #3 (Remarks to the Author):

The authors have incorporated all my remaining suggestions.

I believe that this manuscript is now a very valuable contribution to the field. As the authors, I am not aware of any other study in which subjects played the finitely repeated prisoner's dilemma over multiple days. It's exactly this interaction over several days that makes this study interesting. The paper shows that even if subjects have sufficiently many interactions to learn the logic of the game, cooperation does not break down (as would be predicted if all subjects were perfectly rational, and if this rationality became common knowledge).

In short, I support publication of this manuscript, and I would like to thank the authors for their efforts.

Response to reviewer comments.

The reviewers did not make any comments to which we needed to respond (see below).

Reviewer #1 decided not to leave any comment, largely retaining their previous opinion on the manuscript.

Reviewer #2 (Remarks to the Author):

I appreciate the effort made by the authors in addressing the points I raised in my last report, and I am now fully satisfied. I am sure that, in this version, it is a very nice and interesting paper, that of course deserves publication in Nature Communications, and I believe it will give rise to many productive discussions.

Reviewer #3 (Remarks to the Author):

The authors have incorporated all my remaining suggestions.

I believe that this manuscript is now a very valuable contribution to the field. As the authors, I am not aware of any other study in which subjects played the finitely repeated prisoner's dilemma over multiple days. It's exactly this interaction over several days that makes this study interesting. The paper shows that even if subjects have sufficiently many interactions to learn the logic of the game, cooperation does not break down (as would be predicted if all subjects were perfectly rational, and if this rationality became common knowledge).

In short, I support publication of this manuscript, and I would like to thank the authors for their efforts.